# In Vitro Evaluation of the Antifungal Activity of *Trigonella foenum-graecum* Seed Extract and Its Potential Application in Plant Protection

**DOI:** 10.3390/plants14213320

**Published:** 2025-10-31

**Authors:** Stelica Cristea, Alina Perisoara, Bianca-Maria Tihauan, Manuela Diana Ene, Mariana Constantin, Alexandru-Mihai Florea, Elena Ştefania Ivan, Relu Cristinel Zala, Bogdan Purcăreanu, Dan Eduard Mihaiescu, Lucia Pirvu

**Affiliations:** 1Plant Pathology Department, University of Agronomical Sciences and Veterinary Medicine, 59 Mărăşti Blvd., District 1, 011464 Bucharest, Romania; stelica.cristea@usamv.ro (S.C.); elena.ivan@qlab.usamv.ro (E.Ş.I.); cristinel.zala@usamv.ro (R.C.Z.); 2Biotehnos SA, 3-5 Gorunului Street, 075100 Otopeni, Romania; diana.ene@biotehnos.com (M.D.E.); alexandru.florea@biotehnos.com (A.-M.F.); bogdanpb89@gmail.com (B.P.); 3Research Institute, University of Bucharest-ICUB, Splaiul Independenței, No. 95, District 5, 050095 Bucharest, Romania; ciubuca.b@gmail.com; 4National Institute for Research & Development in Chemistry and Petrochemistry-ICECHIM, 202 Independentei Spl., 060021 Bucharest, Romania; mariana.constantin@icechim.ro; 5Faculty of Pharmacy, Titu Maiorescu University, 16 Bd. Gh. Șincai, 040441 Bucharest, Romania; 6Department of Science and Engineering of Oxide Materials and Nanomaterials, National University of Science and Technology Politehnica Bucharest, 011061 Bucharest, Romania; 7Department of Organic Chemistry, National University of Science and Technology Politehnica Bucharest, 011061 Bucharest, Romania; dan.mihaiescu@upb.ro; 8National Institute for Chemical-Pharmaceutical Research and Development, District 3, 031299 Bucharest, Romania; lucia.pirvu@yahoo.com

**Keywords:** eco-friendly biofungicides, *Trigonella foenum-graecum* seed extract, secondary metabolites

## Abstract

In the context of promoting ecological alternatives to synthetic pesticides, this study investigates the antifungal activity of *Trigonella foenum-graecum* L. seed extract and its potential application in plant protection. The extract, obtained by maceration in 40% ethanol, was analysed using UV-Vis spectrophotometric methods to assess its phytochemical composition, including phenolic compounds, reducing sugars, and soluble proteins, as well as antioxidant activity in acellular system (ABTS, DPPH, TEAC, and CUPRAC) and CAT, SOD, peroxidase, and lipid peroxidation in planting material lysate. Additionally, the extract was qualitatively analysed using ATR-FT-IR and FT-ICR-MS methods. The antifungal activity was tested in vitro against three fungal strains, revealing significant inhibitory effects, especially on *Fusarium graminearum* and *Monilinia laxa*. Following the biogermination study on wheat seeds, it was highlighted that the extract obtained from fenugreek seeds manifested a strong inhibitory effect, especially at the highest concentration (1.50%) studied, probably due to the high content of phenols and presence of steroidal saponins (diosgenin and precursor diosgenin–protodiosgenin) and pyridine alkaloids (trigonelline). These findings suggest that *Trigonella foenum-graecum* seed extract possesses potent antifungal properties, making it a promising candidate for the development of biofungicides in sustainable agriculture.

## 1. Introduction

Commercial agriculture relies primarily on chemical fungicides to protect crop plants against fungal pathogens by destroying and inhibiting their cells and spores. However, their accessibility and low cost result in overuse or repeated applications [1]. This excessive or abusive use of fungicides has toxic effects on beneficial living systems, human health, animals, and the environment too. Furthermore, the emergence of resistant strains of fungal phytopathogens makes treating fungal plant diseases increasingly difficult. Consequently, the development of healthy, non-toxic and environmentally friendly alternative approaches (green fungal control strategies) to chemical and synthetic fungicides is very useful and necessary in the control of fungal infections of plants [2,3]. The vast majority of known fungal species are strictly saprophytes; only very few species (less than 10% of identified fungi) can colonise plants. Phytopathogenic fungi are associated with an even smaller number of these plant colonisers. However, phytopathogenic fungi are key causal agents among phytopathogens for devastating epidemics of crop plants, in addition to leading to persistent and substantial losses in crop yield annually. Thus, phytopathogenic fungi are sought to be combated by both scientists and growers equally due to these important economic factors [4].

Moreover, in horticultural ecosystems, the pernicious action of phytopathogenic fungi influences the physiological resistance and fecundity of plant varieties. Among the most commonly encountered pathogens are *Fusarium*, *Monilinia*, and *Aspergillus* species [5].

*Fusarium graminearum* (*F. graminearum*), a prevalent fungal pathogen, induces *Fusarium* ear blight or *Gibberella* rot in cereals, causing grain discolouration, mould proliferation, and reduced grain viability, thus favouring significant production losses and product quality degradation [6,7]. In addition, its tendency to produce mycotoxins, especially deoxynivalenol (DON), raises numerous concerns regarding food safety and associated economic repercussions. Effective management strategies require an integrative approach that includes new cultural practices, chemical interventions, and an understanding of genetic resistance mechanisms [8] characteristic of this species. *Monilinia laxa* (*M. laxa*) is one of the main fungal phytopathogens that causes brown rot in stone fruit crops, especially peaches, nectarines, cherries, and apricots [9]. As a necrotrophic pathogen, *Monilinia laxa* secretes cell wall-degrading enzymes such as pectin methyl esterases and a series of phytotoxins, thereby opportunistically colonising fruits both before and after harvest, leading to substantial yield losses and quality degradation. After harvest, its persistence generates spoilage, exacerbating the ramifications of economic losses for growers and distributors. Effective management of this pathogen requires a comprehensive approach that includes improved cultural practices, chemical interventions, and sanitation measures to mitigate disease incidence and maintain orchard productivity [10,11].

The pathogenic potential of the *Aspergillus niger* (*A. niger*) species is manifested especially after harvest, as it colonises harvested crops, exploiting nutrient substrates and environmental conditions favourable for proliferation, degrading enzymatic plant tissues, and causing spoilage. Having the potential to synthesise mycotoxins, it thus compromises both the quality and safety of agricultural stores and commodities, requiring complex management strategies to mitigate economic losses and ensure food security [12].

A viable alternative for plant protection may be the use of biologically active products obtained from plant extracts, due to the multitude of active substances they contain [13]. In particular, the use of primary and secondary metabolites, such as volatile compounds (low-molecular-weight phenols), proteins, carbohydrates, fats, sterols, alkaloids, anthocyanins, anthraquinones, carotenoids, phenolic compounds, compounds containing sulphur molecules (phytoalexins), and aromatic hydrocarbons [14,15,16], which exhibit biological activity against pests, presents notable potential. According to the literature, plant extracts obtained from different parts (roots, seeds, fruits, leaves, flowers, rhizomes, or the whole plant) have been shown to exhibit antimicrobial activity against potential plant pests. This effect is attributed to their composition, which includes a range of phytochemical compounds that can act synergistically to inhibit the development of phytophagous strains [17]. Although studies have demonstrated the inhibitory effects of various plant extracts [15] on pathogenic strains such as *Aspergillus flavus*, *Fusarium oxysporum*, *Aspergillus fumigatus*, *Fusarium verticillioides*, *Penicillium expansum*, and *Penicillium brevicompactum* [18], there is still a limited number of commercially available products for plant protection based on biologically active substances (e.g., cinnamon oil, thyme oil, and citrus seed extract) [19].

Given the rich phytochemical composition of *Trigonella foenum-graecum* L. (*T. foenum-graecum*) seed extract, this study aims to evaluate its potential antifungal effect against three fungal strains (*Aspergillus niger*, *Monilinia laxa*, and *Fusarium graminearum*) with significant economic impact on horticulture and agriculture, to develop a potential natural product used in plant protection. For the following reasons, it is proposed that classic extraction technology of maceration at ambient temperature and thermal assistance (rotary evaporator) in the concentration phase, respectively, should be used, alongside the use of solvents from the category of concentrated alcohols (ethanol 70%): (1) the need to preserve the main active compounds with fungicidal effect as volatile compounds and (2) the solubility of the active compounds being the maximum in concentrated alcohols. The present study includes qualitative determinations (ATR-FT-IR and FT-ICR-MS methods) of biological compounds from the *Trigonella foenum-graecum* extract conditioned in 40% ethanol (the solvent concentration was chosen to minimise any potential negative impact on plants), a calculation of the reducing sugar and soluble protein content, and also evaluates the antioxidant properties of the extract in an acellular system using four different assays—CUPRAC, TEAC, DPPH, and ABTS—targeting distinct antioxidant mechanisms and CAT, SOD, peroxidase, and lipid peroxidation in the seedling lysate. The study also assesses the effects of the bioassays on wheat seed germination and highlights the antifungal activities of the extract against *Fusarium graminearum*, *Monilinia laxa*, and *Aspergillus niger*.

## 2. Results and Discussion

### 2.1. Physico-Chemical Screening of Fenugreek Extract in 40% Ethanol

#### 2.1.1. Spectrum Evaluation by ATR-FT-IR

The drying process of the fenugreek seed extract in 40% ethanol resulted in a density of 1.08 ± 0.001 g/mL and a dry matter content of 42%. The qualitative profile of both the extract and the plant material was performed by ATR-FT-IR and is represented in Figure 1. FT-IR assessments provide easy and rapid information on biochemical samples regarding the composition and macromolecular structure [20].

In case of both the extract and plant material, the bands in the range of 4000–500 cm^−1^ highlight the presence of biochemical compounds belonging to the category of carbohydrates, lipids, proteins, and fibres, as well as aromatic compounds. The regions that reveal information about the content of the studied material are the region between the 900–1350 cm^−1^ bands, specific for carbohydrates; the amide I/II region, between the 1450–1650 cm^−1^ bands specific for proteins; the amide III region, specific for the 1230 cm^−1^ band [21,22]; the region between the 2800–3500 bands, specific for the O-H, N-H stretching vibrations; and the C skeleton fingerprint vibrations are attributed to lipids [23]. The bands 3330.37 cm^−1^ and 328.52 cm^−1^ are attributed to the O-H vibrational stretching, which reveals the presence of starch fibres and the N-H vibrational stretching, specific to protein amide A [24]. The bands 2971.58, 2924.23, 2927.08, and 2853.83 cm^−1^ are attributed to the alkyl C-H vibrational stretching. In the case of plant material (fenugreek seeds), the unique bands of 1743.85 cm^−1^, attributed to the presence of saturated fatty aldehyde (C=O vibrational stretching, specific to lipids) [25], and 1237.58 cm^−1^, which signals the presence of soluble fibres (pectin), were removed following the extraction process. The bands 1636.97, 1641.87, 1611.18 (C=O vibrational stretching, amide I), 1577.04, and 1541.37 cm^−1^ (N-H vibrational bending, amide II) indicate the presence of carbohydrates both in the extract and in the plant material, as well as the presence of phenolic compounds. The bands 1376.44–1377.64 cm^−1^ correspond to symmetric and asymmetric CH_3_ bending [26]. The band in the region 1237.58–1042 cm^−1^, due to C-O and C-OH stretching, reveals the presence of carbohydrates, phenolic compounds, and terpene compounds, but also the presence of starch, with a maximum absorption at 1042.60 cm^−1^. The unique band at 873.95 cm^−1^ present in the extract indicates the presence of aromatic compounds (carboxylic acids), such as gallic, chlorogenic, coumaric, ferulic, sinapic, vanillic, syringic, caffeic, and cinnamic acids [27].

Several studies [28,29,30] reveal that ethyl alcohol is effective for the extraction of phenolic compounds, as well as their glycosides, but also phytochemical compounds belonging to the category of terpenoids, sterols, and alkaloids, while water contributes to the extraction of compounds belonging to the category of amino acids, organic acids, and carbohydrates, compounds also found in the *T. foenum-graecum* seed extract studied.

#### 2.1.2. TPC and TFC Content Dosage

Phenolic acids are aromatic secondary plant metabolites associated with an array of functions, such as enzyme activity, structural integrity, nutrient uptake, and protein synthesis [31]. Flavonoids, too, are a type of aromatic secondary plant metabolites, whose role is to provide colour to the flower to attract pollinators, and in leaves, it is believed they confer protection against fungal pathogens and UV radiation and are involved in several growth regulation mechanisms [31]. For centuries, these two plant metabolites have proven themselves effective at combating different illnesses [31]. Following the study conducted on the *T. foenum-graecum* seed extract conditioned in 40% ethanol, it was highlighted that it has a high concentration of polyphenols and flavonoids (Table 1), with a content of 10.733% flavonoids from the total amount of quantified phenolic compounds. The total polyphenol content expressed as caffeic acid was determined by the Folin–Ciocalteu method and was found to be 36.817 ± 0.65 mg/mL, a significantly higher concentration than that found by Rahmani et al. [32] on four types of fenugreek extract (of different varieties) obtained in 96% methanol, which ranged from 1613 to 2083 GAE mg/g extract. It is likely that the type of solvent and the higher concentration of water used in the extraction system led to a higher content of phenolic compounds being obtained. At the same time, the extraction time can be an important factor in the extraction of phytochemical compounds, and this was highlighted in the study conducted by Turker et al. [33], obtaining a higher concentration of polyphenols (17.51 ± 0.28–18.30 ± 0.26 GAE mg/g dry extract) after 8–12 h of maceration of fenugreek seeds in absolute ethanol. Our results are in agreement with those obtained by Lohvina et al. [34], where the highest concentration of phenolic compounds was obtained in fenugreek seed extracts in 70% ethanol (from different varieties) compared to the extract variants in 30%, 50%, and 96% ethanol.

The flavonoid content expressed in rutin was determined by the spectrophotometric method, with a concentration of 3.9517 ± 0.007. Jokic et al. [35], in the study on the extract of some variants of soybean (*Glycine max*) in ethanol obtained by maceration at different concentrations (50, 60, 70, and 80%) and temperatures (25, 40, 50, 60, 70, and 80 °C), obtained a lower yield of flavonoid compounds (2.56 ± 0.04–0.60 ± 0.03 mgCE/gdb catechin equivalents on dry soybean basis). The highest value was found in the case of the variant obtained in 50% ethanol at a temperature of 80 °C. However, in the study conducted by Norziah et al. [36], it was highlighted that the aqueous extract of germinated fenugreek seeds recorded a better yield regarding the content of flavonoids (38.5 mg CE/g) compared to the extract variants in 75% ethanol and methanol. Ethanol and methanol are solvents with a lower polarity compared to water, and their use as the sole extraction solvent may make them less efficient in the extraction of phenolic compounds [25]. Additionally, the use of water (a polar solvent) in the extraction system may contribute to increasing their extraction yield. Polyphenols and flavonoids have different degrees of polarity [37]. Selecting an extraction solvent system with a polarity as close as possible to the phytochemical compounds to be extracted can ensure an efficient extraction [38]. At the same time, according to Rivas-García et al. [39], the extraction of polyphenols is more efficient when polar extraction solvents are used because the latter interact with polar functional groups. Fenugreek (*T. foenum-graecum*) seeds have a rich content of phytochemical compounds, and their most efficient extraction depends largely on the type of solvents used and the extraction technology applied [40]. Water–ethyl alcohol solvent mixtures exhibit higher polarity than solvents used in absolute concentration, leading to better extraction efficiency of phytochemical compounds [41,42].

In most cases, it is possible to identify organic compounds using only the obtained mass spectrum, due to the FT-ICR-MS ultra-high resolution and precision of the mass-to-charge ratio (*m*/*z*). Based on the molecular formula, ESI+ (1M + nH), the mass spectra of the bioactive compounds were generated using Bruker Compass Data Analysis software, which enabled their identification based on the isotopic pattern in the recorded spectra. According to the results obtained and shown in Table 2, compounds belonging to the categories of pyridine alkaloids (trigonelline), steroidal saponins (diosgenin Appendix A, proto-diosgenin—precursor of diosgenin Appendix A, tigogenin, and yamogenin), and phenolic compounds (vitexin, apigenin-7-O-glucoside, apigenin 6-C-galactoside 8-C-arabinoside, salicylic acid and luteolin-7-O-glucosides, rutin, abscisic acid, and t-resveratrol) were identified, compounds that are known for their antimicrobial, antioxidant [43], and anti-inflammatory properties. Contrary to the results of our study, Sakhira et al. [44] identified a larger number of phenolic compounds, in which the most important are catechin, epicatechin, vanillic acid, caffeic acid, coumaric acid, and cinnamic acid. Pasha et al. [45], regarding the study on two variants of fenugreek seed extract in different solvents (methanol, ethanol), highlights the fact that the methanol-rich extract had a higher composition in phenolic compounds (gallic acid, chlorogenic acid, p-coumaric acid, ferulic acid, sinapic acid, quercetin) compared to the ethanolic extract (gallic acid, vanillic acid, syringic acid, quercetin). Sigh et al. (2020) [46] conducted a study to identify the bioactive compounds present in Fenugreek seed extract using different LC-MS methods. Thus, five bioactive compounds (vitexin, isovitexin, trigonelline, isoorientin, and orientin) were identified using HPLC, and 25 compounds were identified using HPLC-ESI-QTOF-MS/MS based on retention time and molecular mass. Additionally, using fragment models of certain compounds, the following biological compounds were identified and quantified using UHPLC-ESI-MS/MS in multiple reaction monitoring (MRM) mode: pinitol, trigonelline, 4-hydroxyisoleucine, isoorientin, and isovitexin. As evidenced by numerous studies, saponins and phenolic compounds, especially flavonoids, exhibit a range of biological activities, including antiparasitic, anti-inflammatory, antimicrobial, allelopathic, and insecticidal properties [47]. Both benzoic acid and salicylic acid play an important role in the plant resistance process [48]. Hernandez-Guzman et al. (2024) [49], following their study of bioconversion by glycosylation and hydroxylation of diosgenin by the oleaginous yeast *Yarrowia lipolytica* P01a, aimed to improve the antioxidant, antifungal, and antiherbicidal performances of the obtained extracts. Thus, high amounts of protodiosgenin and soyagenin were obtained and they managed to highlight superior results regarding antioxidant activity (up to 97.02% of ABTS radicals and a 33.30% inhibition of DPPH radicals at 1000 mg L^−1^ of diosgenin) and, in subsequent antifungal studies, it had strong inhibitory effects on the strains *Botrytis cinerea* (67.34%), *Alternaria* sp. (35.63%), and *Aspergillus niger* (65.53%). At the same time, the bioconverted extracts also showed more herbicidal activity than to commercially available herbicide products. According to specialised studies [50], aqueous extracts of *T. foenum-graecum* obtained from different parts (seeds, stem cells, leaves, roots) have shown antifungal activity on strains of *Fusarium graminearum*, *Pythium aphanidermatum*, *Botrytis cinerea*, *Rhizoctonia solani*, and *Alternaria* sp. These results may make it a potential product for use in plant protection.

#### 2.1.3. Dosage of Soluble Carbohydrates and Proteins

Carbohydrates play an important role in plant growth and development, fulfilling nutrient functions, being at the same time a source of energy and carbon, but also as a central signalling or regulatory molecule that modulates the expression of genes related to metabolism (phytohormone), stress response, and resistance to phytophage agents [51,52,53]. Qualitative evidence of the presence of carbohydrate components in *T. foenum-graecum* seed extract was obtained by quantifying the content of reducing sugar. From the data in Table 3, it can be seen that using the DNS and Bradford method, a high concentration of reducing carbohydrates was obtained: 19.20 ± 0.35 D-glucose mg/mL and soluble proteins 1.96 ± 0.07 BSA mg/mL. This confirms the fact that the extract studied can function as a good nutrient and antioxidant for plant cells. The study by Dawood et al. [52], regarding the effect of foliar application of four types of aqueous fenugreek extracts of different concentrations (5%, 10%, and 15%) on two wheat varieties (Gimeza and Sakha), highlighted the fact that these had beneficial effects not only on the growth and development parameters of wheat seeds, but also on the yield of wheat grains and their biochemical constituents. The study conducted by Patel et al. [54] on 13 genotypes of fenugreek leaves indicates a soluble protein content ranging from 1.82 to 0.99% and a reducing sugar content ranging from 1.59 to 0.73%. Ghevariya et al. [55], following a comparative study of bioactive compounds found in leaves, stem cells, seeds, and microplants/shoots of *Trigonella foenum-graecum*, show that the highest content of reducing sugar was found in the seeds, and fenugreek leaves recorded the highest total protein content.

### 2.2. Determination of Antioxidant Activity

Free radicals, and more specifically ROS (reactive oxygen species) and RNS (Reactive Nitrogen species), are molecules highly dangerous to living cells, damaging DNA, proteins, and membrane lipids [56]. They include both free and non-free radical intermediates, such as hydroxyl radicals (OH^•^), hydrogen peroxide (H_2_O_2_), or peroxynitrite (ONOO^−^) [57]. The three primary species of ROS are O_2_^•−^, H_2_O_2_, and HO^•^. O_2_^•−^ is produced enzymatically and is transformed by superoxide dismutase (SOD) into the less toxic H_2_O_2_. O_2_^•−^ and H_2_O_2_ can react together to generate HO^•^ [58], the most reactive ROS. O_2_^•−^ can also react with ^•^NO to form ONOO^−^ [57]. As a matter of fact, mitochondria are both the main endogenous source of ROS [59] (bringing about up to 90% of the total ROS in the cell), which can be generated at the level of the electron transport chain, and a major target of these same molecules [60]. Up to 2% of the oxygen consumed during cellular respiration is turned into ROS [59]. Among the deleterious effects they have on mitochondria, or the cells in general, are mutations and deletions of the DNA, peroxidation of membrane lipids, and permeabilisation of the membrane due to the opening of the mitochondrial permeability pore [56]. For this reason, the cells need to regulate ROS and RNS concentrations tightly. An imbalance between these species and the cell’s ability to detoxify them, via several specific enzymes such as SOD, Catalase, and Glutathione Peroxidase, or to repair the damage caused by them, is referred to as oxidative stress [61,62,63,64,65]. Phytochemicals present in plant extracts exhibit antimicrobial and antioxidant activity, and this is largely due to the presence of phenolic compounds [66]. According to our results, the fenugreek extract conditioned in 40% ethanol possessed a high concentration of phenolic compounds. In the present study, antioxidant activity was determined by four different methods, two of which, DPPH and ABTS, are methods based on the free radical scavenging mechanism, and the CUPRAC and FRAP methods are methods based on the reduction mechanism of ions (copper and iron). The CUPRAC method, unlike the FRAP method, is a less used and relatively new one, in which neocuproin is used as an oxidising agent [67]. In most studies, the antioxidant activity of plants is determined by the classical DPPH and ABTS methods. The protons of free radicals present in DPPH and ABTS reagents decrease rapidly in speed at the moment they come into contact with protons of radical scavengers [68]. Due to the high sensitivity of the ABTS method, the reaction kinetics are faster, and a large part of the phytochemical compounds show better antioxidant activity by this method, compared to the DPPH method [69]. At the same time, the ABTS radical cation shows reactivity with most antioxidants soluble in both organic and polar solvents; thus, lipophilic and hydrophilic antioxidants can be identified [56]. In the study conducted by Kaya and Akbaş [70] on the determination of the antioxidant activity of four variants of *Peganum harmala* seed extract, obtained in different solvents (water, ethanol, methanol and chloroform), it was highlighted that the best antioxidant activity was obtained in the case of the aqueous extract by applying the CUPRAC method (0.714), the methanolic extract showed better antioxidant activity by the DPPH method (74.06%), and the ethanolic extract by the ABTS method (72.06%). Wasim Bari et al. [67], by studying the antioxidant activity of the methanolic leaves extract of *Sphagneticola calendulacea* (L.) Pruski by five different methods (FRAP, CUPRAC, DPPH, ABTS, and nitric oxide free radical scavenging method) obtained better antioxidant activity by applying the ABTS (89.63%) and DPPH (79.43%) methods. Bukhari et al. (2025) [71] studied the antioxidant activity of four types of extracts obtained from fenugreek seeds in different solvents (methanol, ethanol, dichloromethane, acetone, hexane, and ethyl acetate) using the DPPH and ABTS methods, where they highlight the fact that all six types of extracts obtained showed a strong antioxidant effect, especially the extracts obtained in polar solvents, ethanol and methanol. The strongest antioxidant effect was recorded in the case of the extract in ethanol in both methods studied. This is due to the high concentration of phenolic compounds found in the composition of the extract.

In our experiment, the strongest antioxidant activity of the *T. foenum-graecum* seed extract studied was demonstrated by the DPPH method (Table 4), followed by ABTS, and the weakest antioxidant activity was recorded by the CUPRAC method. These results are in line with the trend of the majority of studies, indicating that ABTS is the method that best reveals the antioxidant activity of fenugreek extracts, followed by DPPH, while FRAP and CUPRAC show results with high variability but significance in some studies. Antioxidant activity tests (DPPH, ABTS, FRAP, CUPRAC) have an important biological significance in treatments with plant extracts to stimulate germination processes, as they highlight their ability to protect seeds against oxidative stress, which can inhibit germination of plant germplasm. The extract analysed in the present study, by its high antioxidant activity, as evidenced by the DPPH assay, suggests that the composition of the extract may protect the seeds from oxidative stress, which could inhibit root development or lead to slower germination. The high reducing potency demonstrated in the FRAP assay suggests that it also intervenes in the processes that maintain the cellular integrity of the seed germinating material during germination, protecting them from damage to membranes or other essential components that would hinder normal development. The results obtained from the CUPRAC test suggest a potential protective effect against potential damage caused by the presence of heavy metals or other contaminants in the soil, thus stimulating germination. All these experimental observations highlight the ability of the analysed extract with high antioxidant activity to help the planting material to better cope with adverse environmental conditions and may stimulate faster and healthier germination. According to our results, fenugreek seed extract can ensure the natural health state (homeostasis) of reactive oxygen species. These can act as signalling molecules with a role in plant growth and development [70].

### 2.3. The Effect of the Trigonella foenum-graecum Seeds Extract and the Solvent on Wheat Seeds (Triticum Aestivum)

A product/active substance with potential use in the plant protection and phytostimulation industry must be tested to determine possible toxic or phytostimulant effects that can manifest on plants [72]. The most convenient and widely used test for excess salinity or the presence of allelopathic substances (e.g., phenolic compounds) in plant extracts or compost is the seed germination bioassay [73,74,75].

According to the specialised literature [75,76,77], seeds can take up from the treatment medium substances that can fulfil a nutritional or allelopathic role, which enter into their composition, and this can be evidenced in the growth and development of their roots. According to Emino and Warman [78], the effect of the new substances/extracts studied on plants is reflected in the Gi value obtained. This, in turn, is obtained by quantifying the values of the germination index (RSG) and the relative root growth index (RRG) [76].

In the present study, the effect of *Trigonella foenum-graecum* seed extract at different concentrations (0.10%, 0.50%, 1.00%, and 1.50%) on wheat seeds was compared with its solvent (40% ethanol). This was performed to highlight the potential stimulant or phytotoxic effect of the extract on the treated seeds. Germination percentage (GP%) was determined by reporting the number of germinated seeds in the sample/control with the number of seeds studied [79]. GP indicates possible allelopathic effects on the studied seeds [80,81]. However, it is an indicator that cannot predict the possible delayed seed germination that could be caused by the presence of phytochemical compounds present in the plant extract. As can be seen from Table 5, the GP (%) values for both solvent and extract were in the range of 73.33–91.67%. It is noteworthy that the extract had a positive impact on wheat seeds compared to the solvent, except for the concentration of 1.50%, where the GP (%) value for the solvent (90 ± 2.47%) was higher compared to *Trigonella foenum-graecum* extract (85 ± 0.66%). Mominul Islam et al. [82] studied the phytotoxic activity of methanol obtained from *Ocimum tenuiflorum* (*Lamiaceae*) on six seed varieties (*Lepidium sativum*, *Medicago sativa*, *Lolium multiflorum*, *Echinochloa crusgalli*, *Phleum pratense*, *and Phleum pratense*) in different concentrations (3, 10, 30, and 100 mg extract equivalent mL^−1^), highlighting that the germination percentage was significantly reduced with increasing extract concentration studied, for all seed varieties, except lettuce and yard grass seeds. Azizi et al. (2011) [83] conducted a study in which the effect of four aqueous extracts obtained from different parts of *Trigonella foenum-graecum* (leaf, stem, seed, pod, and whole plant) on four seed varieties (*Glycine max*, *Sesamus indicum*, *Amaranthus retroflexusI*, and *Abotilon theophrasti*) was monitored, where it was demonstrated that all the extract variants obtained presented inhibitory effects on the seeds studied. As our study reveals, there is a correlation between the germination percentage and the extract concentration studied. The most inhibitory effect was recorded in the case of the extracts obtained from the *Trigonella foenum-graecum* seeds and leaf.

Regarding the relative germination index RSG (%), it was interpreted by reporting the average of germinated seeds in the sample/solvent compared to the average of germinated seeds in the control sample (distilled water). The extract sample’s RSG values (%) fell within the 94.80–87.93% range, and for the solvent, they fell within the range of 98.21–78.57% (Table 6). It is worth noting that the values of the relative germination index in the case of the extract decrease directly proportionally with the increase in the extract dose (non-significant statistically, *p*-value > 0.05). The same trend was also observed in the case of the solvent, except for the maximum tested concentration (96.42%), where the RSG value is higher than compared to the values obtained at the concentrations of 0.50% and 1.00%, but also than that recorded in the case of the extract (87.93%), at the maximum tested concentration (non-significant statistically, *p*-value > 0.05). It can be observed that the fenugreek seed extract conditioned in 40% ethanol at higher concentrations may have a slight inhibitory effect due to the higher content of phenolic compounds or residual solvents left after the extraction process. There are a series of studies [84] that have demonstrated the allelopathic effect manifested by the phenolic compounds present in plant extracts on the development and growth of plants, mainly intervening in physiological processes, such as the absorption of nutritional compounds, protein synthesis, plant cell permeability, respiration, and photosynthesis, as well as the water ratio. Our study can also be correlated with the results obtained by Xuan et al. [85], where it emerged that the aqueous extract of neem had inhibitory effects on the six seed varieties studied: alfalfa, beans, carrots, radishes, rice, and sesame.

The relative root growth index (%) is interpreted by comparing the average root length of germinated seeds from the extract/solvent sample to the average root length of the control sample. Data from the literature [86] reveal that the RRG index is a more sensitive indicator of the toxicity of biologically active compounds compared to RSG. One reason for this could be that roots can easily absorb the substances present in the treatment medium, and this can be observed in their development and growth [74]. As can be seen in Table 7, the RRG values (%) for the extract were within the range of 85–70–35.70%, and for the solvent between 109.10 and 70.48%. If we compare both the control and the solvent, we can see that the extract has strong allelopathic compounds that affect the growth of wheat seed roots; the greatest impact was seen at the maximum concentration tested (1.50%), where the lowest RRG value was recorded (35.79%, statistically significant *p* < 0.05, concerning both the solvent and the control sample, Figure 2). This result also correlates with the data obtained regarding RSG and GP from our study. Roy et al. [87] investigated the effect of aqueous extracts obtained from different parts of the banana plant (rhizome, root, leaf sheath, and leaf blade) at 15%, 25%, 50%, and 100% concentrations on six varieties of seeds (lettuce, red amaranth, amaranth, radishes, cucumbers, gourd, beans, and okra), where a strong inhibitory effect on root development, as well as on the germination capacity of seeds, was demonstrated for all extract variants, especially in the case of the extract obtained from rhizomes at the maximum concentration tested (100%).

Interpretation of the results regarding the germination index (Gi% %) is based on the results obtained for the RSG index and the RRG index [80]. According to Cesaro et al. [88], the germination index is a parameter included in the list of quality assurance regulations in the Italian legislation on the marketing of composts with a role in plant protection and phytostimulation. Following the results obtained regarding the Gi index, we can interpret whether or not the fenugreek extract has a phytostimulating or phytotoxic effect on wheat seeds. The data obtained show that the *Trigonella foenum-graecum* seed extract exhibits a strong phytotoxic effect at the highest concentration tested (Gi 32.98%) (Figure 3), the Gi value recorded being below 50%. As for the solvent, it showed a slightly stimulatory activity at a concentration of 1.50% (Gi 105.73%), the Gi value being above 100%. Reigosa et al. [89], following their study on the impact of some small molecular size phenolic compounds (gallic acid, *p*-coumaric acid, *p*-vanillin, *p*-hydroxybenzoic acid, ferulic acid, and vanillic acid) on six weed varieties (*Chenopodium album* L., *Plantago lanceolata* L., *Amaranthus retroflexus* L., *Solanum nigrum* L., *Cirsium* sp., and *Rumex crispus* L.), demonstrated that they exhibited inhibitory effects only at the highest concentrations studied, but at low concentrations, they had no effect, and in some cases they exhibited stimulatory effects.

Our results are also correlated with those obtained by Dai et al. [90], who investigated the effect of the aqueous extract of *Falveria bidentis* leaves on three plant varieties (Shanghai green, barnyard grass, and wheat), demonstrating that the inhibitory response of the extract was stronger at higher concentrations. It displayed inhibitory effects on both the germination rate and the development of plant roots. As we have mentioned before, phenolic compounds, used at certain concentrations, can exhibit undesirable effects on plants, and these can manifest themselves at molecular, physiological, and biochemical levels [91]. They can induce oxidative stress as a consequence of the high production of ROS (reactive oxygen species). It is known that ROS manifests itself as a signalling molecule, intervening in the process of regulating the response of plants to abiotic and biotic stress factors. At the same time, ROS and phytohormones are involved in plant growth and development. Abscisic acid and ethylene are considered stress hormones that are involved in the process of seed germination, especially dormancy [92].

The germination process is a crucial stage in the life cycle of plants, where seeds start to germinate and develop into new plants, and is a process that is highly sensitive to various environmental factors, with water deficit or drought stress being the most important stress factors. Seed germination and the post-germination phase are finely regulated by oxidative balance, a process involving lipid peroxidation, reactive oxygen species (ROS) formation, and enzyme activity, all of which have a direct impact on plant viability and subsequent plant development [93,94,95]. Polyunsaturated fatty acids, the predominant molecules in the membrane structures of plant cells, are vulnerable to ROS attack, and their peroxidation affects both germination capacity and seed longevity [88,89,90]. One of the most common forms of ROS is hydrogen peroxide, which has a significant cytotoxic effect on cells, being involved in oxidative degradative processes of cell membranes and disrupting essential physiological processes during germination [90,93,94]. In the early stages of imbibition, lipid peroxidation processes are activated and cell membranes undergo structural damage, leading to the accumulation of malondialdehyde (MDA), commonly used as a marker of oxidative stress [96]. The increase in MDA concentration is directly associated with enhanced cellular degradation and abiotic stress, in particular water stress, and this phenomenon contributes to a decrease in germination rate and a progressive reduction in seed viability. Due to all of these aspects, the accumulation of MDA serves as an indicator of irreversible damage to cellular structures, leading to a decrease in germination potential and a shorter seed life [93,94,95]. The stability of cell membranes is directly influenced by the intensity of the lipid peroxidation process, and the variations in MDA levels observed after seed pretreatment can be correlated with the activity of antioxidant enzymes involved in free radical detoxification, such as superoxide dismutase (SOD), peroxidase (POD), and catalase (CAT) [97,98]. These enzymes play a crucial role in maintaining cellular redox balance, contributing to ROS neutralisation and protecting the structural integrity of cells [95,96,97]. Under conditions of water stress, plants are forced to develop protective mechanisms to counteract the harmful effects of the reactive oxygen species generated, and antioxidant enzymes such as catalase (CAT), superoxide dismutase (SOD), and peroxidase (POX) play a central role in this protective mechanism. These enzymes, through their specific catalytic properties (SOD—converts superoxide anion into hydrogen peroxide, thus reducing the oxidative potential at the cellular level; CAT—catalyses the decomposition of hydrogen peroxide into water and oxygen, protecting cells from its cytotoxic effects; peroxidase—in the presence of electron donors breaks down hydrogen peroxide, increasing the protection of cellular structures by preventing its accumulation in toxic concentrations), form an antioxidant defence system that maintains the oxidative balance and protects seeds in the critical phases of imbibition and germination, preventing catabolic processes at cellular level and supporting normal plant development [99,100,101]. Recent studies suggest that modulating the oxidative balance, through targeted interventions that limit ROS accumulation and lipid peroxidation, can significantly improve germination capacity and seed longevity, offering new prospects for improving agricultural productivity and crop sustainability under abiotic stress. In taking all of this into account, an extended experimental study was developed to evaluate the antioxidant properties of fenugreek seed extract in the seedling material based on its ability to modulate the enzymatic activity of protein enzymes involved in phase I, counteracting oxidative stress. The experimental results are presented in the figures below (Figure 4).

From the experimental data, it is observed that the activity of antioxidant enzymes (CAT, SOD, peroxidase) and the concentration of the stress marker (MDA) in *Triticum aestivum* are up-regulated and significantly increased under the influence of abiotic stresses, indicating the activation of the plant defence mechanisms. Under normal conditions, these values are relatively low, while under stress conditions (water, heat, etc.), the values increase to combat lipid peroxidation and ROS accumulation. The process is greatly accelerated by the presence of fenugreek seed extract, which manages under water stress conditions to counteract the harmful effects of reactive oxygen species by activating protective enzymatic mechanisms.

### 2.4. Trigonella foenum-graecum Seed Extract Antifungal Activity

Following the qualitative evaluation of the antifungal activity of the standardised extract in 5 mg/mL gallic acid (GAE) obtained from *Trigonella foenum-graecum* L. (Fenugreek) seeds in 40% ethanol, it can be observed that it is effective, especially at the concentration of 500 mg/mL, on the three tested strains. The extract concentrations (62.5–500 mg/mL) were selected to cover a broad range for in vitro bioactivity screening and to identify the minimum inhibitory concentration (MIC) capable of producing measurable antifungal effects. These concentrations were chosen based on preliminary solubility and diffusion assays, as well as values commonly reported in similar in vitro studies involving crude plant extracts. It is important to note that these values do not directly reflect agronomically applicable doses but rather serve as an initial step to characterise the antifungal potential of *Trigonella foenum-graecum* extract. Further investigations under greenhouse and field conditions are needed to calibrate these concentrations for practical application in crop protection.

According to the results obtained, it can be observed that the most pronounced fungicidal effect was obtained on the *Fusarium graminearum* strain, where the concentration of 500 mg/mL inhibited the development of the strain by 84.43 ± 4.29% compared to the antifungal control Amphotericin B (10 µg/mL) (Table 8, Figure 5b) and with the solvent (64.56 ± 5.88%). For the next three concentrations tested (250 mg/mL, 125 mg/mL, and 62.5 mg/mL), lower inhibitory values were obtained, the associated effect being possibly fungistatic. However, it is important to note that at these concentrations, 40% ethanol was devoid of antifungal activity; the results obtained are entirely associated with the effect of the extract. In the case of the *Aspergillus niger* strain, the most effective antifungal activity was observed at 500 mg/mL, namely 78.01 ± 5.61%, while at 250 mg/mL, 61.78 ± 3.72% inhibition was obtained compared to the Nystatin control (5 µg/mL) (Table 8, Figure 5a). However, the antifungal effect of 40% ethanol was more pronounced compared with the extract at the tested concentrations; the 500 mg/mL sample, where the inhibition was 87.32 ± 5.74% was the significant one. The following concentrations tested in the presence of the extract indicated significantly lower inhibition values, respectively, 37.80 ± 1.54% and 22.62 ± 3.11. In the case of the *Monilinia laxa* strain, the 500 mg/mL concentration of the extract led to an inhibition of 66.23 ± 5.55% compared to the Imidazole control (10 µg/mL) (Table 8, Figure 5c). It can be observed for the tested strains that the activity of the extract is directly proportional to the tested concentrations. The 40% ethanol showed significant percentages of inhibition for tested concentrations, but a synergistic/potentiating effect of the extract activity can be observed, especially in the *Monilinia laxa* strain.

According to the literature, a series of phytochemical constituents or main compounds identified in *Trigonella foenum-graecum* L., namely saponins, steroidal saponins, flavonoids, phenols, proteins (especially in aqueous and ethanolic extracts), carbohydrates, or alkaloids (especially in aqueous extracts), are responsible for the antifungal activity on *Aspergillus niger* strains [102,103]. As for the effect of Fenugreek extract on the *Fusarium graminearum* strain, the results obtained correspond to those reported in the literature [50], in particular, the presence of a dose–response effect on the antifungal activity. As for *Monilinia laxa*, the literature does not provide results that directly correlate the antifungal effects of Fenugreek extract with this pathogen. Most studies focus on evaluating the effect of volatile oils on strains of the *Monilinia* species [104]; thus, the determinations made in the present study are innovative in terms of the development of new biopesticides necessary to maintain crop safety. After harvest, the biggest challenge facing the agro-tech market is certainly fungal. Fungi, without effective control methods, can result in harvest losses of up to 24% [105]. Most of the losses are due to diseases of commercially important fruits and vegetables that result from pre- or post-harvest infections with fungal pathogens. This is due to their adaptability, which allows them to grow and develop under storage conditions. Thus, the major fungal threats after harvest are moulds and fungi that can infect a wide range of plant species [106].

The quantitative evaluation of antifungal activity was performed at four extract concentrations for each strain, and the minimum inhibitory concentration (MIC) was thus evaluated, noting that in the case of *Fusarium graminearum* (Figure 6b), the reduction in cell viability reported for the solvent was statistically significant (*p* < 0.5) for all concentrations tested. The most significant reduction (viability 15.57 ± 4.28%) was associated with the concentration of 500 mg/mL extract. In the case of *Aspergillus niger* and *Monilinia laxa* strains, the minimum inhibitory concentration was observed at 500 mg/mL, with no statistically significant differences compared to the activity of the solvent (Figure 6a,c). For the antifungal strain *Monilinia laxa*, lower values of cell proliferation were observed compared to the solvent (positive control). This buffering/protective effect may be due to the presence of phenolic compounds that are associated with the reduction in intracellular oxidative stress (hesperidin, vanillic acid, quercetin, and caffeic acid) generated by the solvent [107,108,109,110].

The MIC was defined as having the lowest concentration of *Trigonella foenum-graecum* seed extract that completely inhibited visible fungal growth after incubation for 7 days, compared to the solvent control. Based on the quantitative evaluation of inhibition and viability assays, the MIC values were determined as 500 mg/mL for *Fusarium graminearum* and >500 mg/mL for *Aspergillus niger* and *Monilinia laxa* (Figure 5a–c). The pronounced inhibitory effect at 500 mg/mL on *F. graminearum* (viability 15.57 ± 4.28%) indicates a fungicidal activity, whereas the partial inhibition and subsequent regrowth observed for *A. niger* and *M. laxa* after subculturing onto PDA without extract support a fungistatic effect at this concentration. Similar fungicidal–fungistatic thresholds have been reported for crude plant extracts rich in saponins and phenolics, which cause irreversible membrane disruption at higher concentrations, while lower concentrations only suppress sporulation and hyphal elongation [111]. These observations reinforce the notion of the concentration-dependent mode of action of the fenugreek extract and justify its potential use as a biofungicide prototype.

To evaluate the effect of the minimum inhibitory concentrations obtained for the development capacity of fungal mycelia, *T. foenum-graecum* extract was incorporated into PDA plates at a final concentration of 500 mg/mL. The plates were seeded with fragments of the active growth zone of the fungal culture with a diameter of 6 mm. The plates were incubated at 27 ± 1 °C for 5–7 days. After incubation, the diameters of the developed colonies were measured. The diameters of the colonies treated with extract were compared with those of the colonies grown on the culture medium without the addition of extract (control). Inhibition (%) was calculated using the formula shown above.

The evaluation of the antifungal effect expressed by inhibiting the development capacity of fungal mycelia was carried out in the presence of 500 mg/mL extract and solvent (40% ethanol) at the same concentrations. The results obtained indicate an inhibition of over 50% for the *Fusarium graminearum* strain (60.1 ± 0.9%), while for 40% ethanol, the inhibition was considerably lower (11.1 ± 1.1%), with significant differences compared to the solvent (Figure 7, *p* < 0.0001). In the case of the *Aspergillus niger* strain and *Monilinia laxa*, the inhibition percentages obtained are similar to those of the solvent, the percentages being below 20%, with only the reduction obtained for *Monilinia laxa* by the solvent being significant (16.5 ± 0.5% extract, 9.06 ± 0.96% solvent, *p* < 0.001).

The antifungal activity of *Trigonella foenum-graecum* seed extract can be attributed to the synergistic action of its major phytochemical classes, particularly steroidal saponins and flavonoids. Steroidal saponins such as diosgenin and protodiosgenin are amphiphilic molecules that interact with membrane sterols (e.g., ergosterol), forming complexes that disrupt fungal cell membrane integrity and increase permeability, leading to the leakage of cellular contents and eventual cell lysis [112,113]. This mechanism has been reported in several *Fusarium* and *Aspergillus* species treated with saponin-rich extracts [114,115]. Flavonoids such as luteolin, vitexin, and apigenin derivatives also exert antifungal effects by interfering with ergosterol biosynthesis, inhibiting hyphal growth, and inducing oxidative stress through ROS generation. Moreover, phenolic acids (salicylic, caffeic, and ferulic acids) may act synergistically by altering cell wall rigidity via interactions with chitin and β-glucan components, thereby potentiating the activity of membrane-active saponins [116]. This combination of membrane disruption, enzyme inhibition, and oxidative imbalance likely explains the strong inhibitory effect observed against *Fusarium graminearum* in our assays [117].

The pronounced antifungal effect observed against *Fusarium graminearum* compared to *Aspergillus niger* and *Monilinia laxa* may be related to species-specific differences in cell wall architecture and membrane composition. *F. graminearum* exhibits a less compact chitin–glucan structure and higher ergosterol [118], which can increase susceptibility to membrane-active compounds such as steroidal saponins and phenolic acids present in the extract. In contrast, *A. niger* and *M. laxa* possess more robust cell walls, rich in melanin and cross-linked polysaccharides [119,120], providing greater resistance to oxidative and surfactant-like agents. This structural variability likely contributes to the differential sensitivity observed among the tested fungi. Similar patterns have been reported for plant-derived saponins and flavonoids, which preferentially disrupt the cell membranes of *Fusarium* species while exhibiting limited effects on melanized or spore-forming fungi.

## 3. Materials and Methods

### 3.1. Method of Obtaining the Vegetable Extract and Physicochemical Characterisation

The plant material used consists of dried fenugreek seeds obtained commercially from the company Ceaiul Casei: country of origin, Romania. The fenugreek seeds were passed three times through purified water and dried in an oven at a temperature of 40 °C for 1 h, the process being followed by their grinding. The fenugreek seed extract was obtained by initially macerating them in 70% ethanol for 10 days at ambient temperature (25 °C). The resulting extraction mixture was separated from the vegetable waste using filter paper. The vegetable waste was taken up again, over which 1000 g of 70% ethyl alcohol was added, with the extraction then being continued for 24 h, at room temperature, away from light, and with occasional shaking. This operation was repeated for another 24 h. The extraction ratio was 1:10 (*w*/*v*). The resulting extract was analysed for gallic acid concentration and concentrated by removing the alcohol in a rotary evaporator, followed by resuming the extraction process in 40% ethanol to achieve a gallic acid concentration of at least 5 mg GAE/mL—*m*/*v*. The resulting extract was stored at 40 °C until the related studies on extract analysis were started.

### 3.2. Qualitative Extract Screening

#### 3.2.1. Highlighting the Presence of Phytochemical Compounds in the Dry Extract of Fenugreek Seeds and Plant Material by the ATR-FT-IR Method

The spectra of the dry extract and vegetal material of fenugreek were obtained using Fourier transform infrared spectroscopy (Thermo Scientific, Waltham, MA, USA, Nicolet iS50 FT-IR Spectrometer with automated beamsplitter exchange) in the mid-infrared range, 4000–500 cm^−1^. Approximately 5–10 mg of dry extract/vegetal material was placed on the surface of the diamond crystal of the attenuated total reflective (ATR), and then it was pressed gently and analysed directly. A total of 64 spectra were acquired for background and 64 spectra for each sample, at a resolution of 4 cm^−1^. Detection was performed using a dedicated DLaTGS detector (InfraTec, Dresden, Germany), and KBr as a beamsplitter. The reading of the FT-IR spectra for the extract sample was performed after it had previously been subjected to the drying process in an oven at 30 °C for 72 h [121].

#### 3.2.2. Identification of Bioactive Compounds Using Fourier Transform Ion Cyclotron Resonance High-Resolution Mass Spectrometry (FT-ICR-MS) Analysis

FT-ICR-MS analysis was performed using a high-resolution mass spectrometer equipped with a 15 T superconducting magnet (Solar X-XR, QqqFT-ICR HR, Bruker, Daltonics, Germany). The instrument was calibrated before the analysis using a sodium trifluoroacetate (NaTFA) solution. The sample was prepared by diluting 30 μL of hydroalcoholic fenugreek extract in 30 mL of ultrapure water and adding 10 μL of formic acid. The resulting solution was ultrasonicated for 10 min and introduced into the electrospray ionisation (ESI) source using the direct infusion system at a flow rate of 120 µL/h. The spectra were recorded using positive ionisation in a mass range between 92 and 1500 amu, with 50 scans, with an ion accumulation time of 0.700 s, a time of flight of 0.001 s, and a data acquisition size of 4,194,304. The extract analysis parameters were as follows: capillary voltage of 3700 V, Spray Shield of −450 V, drying gas pressure (N_2_) 3.0 bar at 180 °C, and a flow rate of 1.2 L/min.

#### 3.2.3. Total Polyphenol Content Dosage—TPC

The total polyphenol content was determined following the method described by Singleton et al. [122], with minor modifications. The total phenol content (TPC) assay uses the electron transfer method to detect the presence of antioxidants, rather than the capacity. However, since it uses a redox-type reaction, it is often considered an antioxidant capacity assay. It uses the Folin–Ciocalteu Reactant (FCR, Sigma Aldrich, Darmstadt, Germany, product code F9252) which can be reduced by the phenolate anions formed by phenols in a basic medium. The scientific consensus is that molybdenum (Mo) allows FCR to receive an electron from phenols, thus being reduced from Mo^6+^ to Mo^5+^. During the reaction, the colour shifts from yellow to blue, and it can be followed by measuring the absorbance at 765 nm. The reaction media is composed of a total of 200 μL of methanol and antioxidant extract, to which 2.5 mL of FCR (diluted in a 1:10 ratio with purified water) is added. After 10 min at room temperature, 2 mL of sodium carbonate 7.5% solution is added, and the reaction medium is left at room temperature for 2 h. A calibration curve was produced for concentrations of caffeic acid prepared with methanol, ranging from 25 to 250 μg/mL. For the extract tested, 3 solutions with either 2, 4, or 6 μL of plant extract are prepared in the same way, and are used to find the average antioxidant concentration with the help of the plot formula. The samples were run in triplicate, and then reported as mean and the ±standard deviation (SD) was calculated. The results were expressed in mg caffeic acid equivalents/L extract.

#### 3.2.4. Total Flavonoid Content—TFC

Total Flavonoid Content antioxidant assays are commonly used for plant extracts, as more than half the identified phenolics found in plants are flavonoids. The most common method is based on the aluminium chloride colorimetric assay, where Al (III) acts as a complexing agent and can form yellow-coloured chelates of Al (III)-flavonoids [123]. Flavonoids possess many oxo and hydroxyl groups, which allow them to bind to metals such as Al (III) at a 1:1 ratio. This reaction can be followed by spectrophotometry, as the absorbance measured at 415 nm increases proportionally to the quantity of chelates formed, and, by extension, proportionally to the quantity of antioxidants in the solution. First, a solution of AlCl_3_ (10%) was made by dissolving 0.9 g of AlCl_3_·H_2_O in 5 mL of H_2_O. This solution was used as the reactant for this test. A 1M potassium acetate solution was also made by dissolving 0.591 g of potassium acetate in 5 mL of H_2_O. A calibration curve was made with 2.8 mL H_2_O, 100 μL of AlCl_3_ (10%), 100 μL of potassium acetate (1 M), and 1500 μL of a mixture of methanol and increasing volumes of rutin (1 mg/mL). The blank contained 2.8 mL H_2_O, 100 μL of AlCl_3_ (10%), 100 μL of potassium acetate (1 M), and 1500 μL of methanol. For the plant extracts tested, 3 solutions were made with the same composition, swapping the rutin from the calibration curve with 2, 4, and 6 μL of plant extract. After leaving the reaction medium at room temperature for 30 min, the absorbance at 415 nm was measured. The calibration curve was made, and the plot line formula was used to calculate the flavonoid concentration in each extract.

#### 3.2.5. Determination of Reducing Sugar Content (RSC)

For the quantification of reducing sugar content, the 3,5-dinitrosalicylic acid (DNSA) method was used, described by Khatri et al. [51], with some modifications. DNSA was prepared using 5.35 g NaOH, 45.5 g Na and K tartrate, and 1.25 g Na bisulfite, dissolved in distilled water. The resulting mixture was heated to 80 °C for 5 min. Measures of 1.25 g phenol and 1.575 g 3,5-dinitro salicylic acid were added and distilled water was added to make a total volume of 250 mL. For sample preparation, 350 µL of the extract sample was taken, and 650 µL of DNSA reagent was pipetted onto it. The mixture was heated to 90 °C for 15 min. After cooling, 150 μL of the reaction mixture was pipetted into 96-well plates, and the absorbance of the resulting solution was read at 540 nm. In parallel, control samples were also prepared, using 40% ethanol. The calibration curve was represented by D-glucose in different concentrations from 0 to 0.1 mg/mL.

#### 3.2.6. Dosage of Soluble Protein Content

The method used in the determination of soluble proteins was adapted from that described by Bradford et al. [124]. Thus, the supernatant was used, and a volume of 10 µL of the extract sample was taken to which 200 µL of Bradford reagent was added. The sample and standard solutions, together with the reagent, were incubated for 15 min at room temperature. The absorbance was read at λ = 595 nm. The calibration curve was made, starting from a stock solution of bovine serum albumin of 2 mg/mL for concentrations that varied between 1.50 and 0.00 mg/mL.

### 3.3. Antioxidant Activity

The benefits of antioxidants have become more mainstream, and more and more studies focus on them. Between 1993 and 2003 alone, the number of publications on this subject quadrupled, and this upward trend has continued in recent decades [125]. Several antioxidant assays can be used to determine the antioxidant capacity of a sample, and they are usually based on electron transfer (ET), hydrogen atom transfer (HAT), or a mixture of both [126,127]. ET-based assays determine the capacity of an antioxidant to transfer one of its electrons and reduce the oxidant. The latter changes colour proportionally to the amount of antioxidant found in the sample, thus making it possible to follow the reaction by measuring the absorbance at a specific wavelength. The majority of HAT assays, on the other hand, involve a competitive reaction between the oxidant and the antioxidant, competing with the substrate [128,129].

#### 3.3.1. Antioxidant Activity by DPPH Method

The DPPH (2,2-Diphenyl-1-picrylhydrazyl Radical) assay is one of the most used mixed-mode methods for determining antioxidant capacity [130]. It combines HAT as well as ET methods. However, it could be considered as mostly an ET method, given the fact that the abstraction of the hydrogen atom by the DPPH^•^ free radical from the antioxidant compound (HAT mechanism) happens less easily than the transfer of the electron from the free radical DPPH^•^ towards the antioxidant compound (ET mechanism). During this assay, a solution of DPPH^•^ free radicals is made by dissolving DPPH, commercially available as a powder, into distilled water. Because the chemical structure allows the decolouration of the free radical via mesomeric effects, the free radicals are highly stable and absorb light in the visible spectrum, giving the solution a deep purple colour. This technique has the advantage of being a simple and fast method to find the antioxidant capacity of a solution, and the reactive being stable enough to be commercialised and ready for use. It does, however, require the use of alcoholic solutions as media, which hinders its use for emulsions, as proteins often precipitate in alcohol. Moreover, due to steric hindrance, large antioxidant molecules react more slowly or not at all, not reaching the radical site [131,132]. Methanolic solutions were made using aliquots of 12 different plant extracts and reaching a total volume of 3 mL. The reaction is started by the addition of 1.5 mL of DPPH reactive (0.04%). It is left for 20 min at room temperature and is followed by measuring the decrease in absorbance at 515–528 nm for the 7 probes with increasing extract volume, as well as for the reference, in triplicate (containing only 3 mL of methanolic solution and 1.5 mL of reactive). For each probe, the inhibition percentage is calculated using the following formula:(1)$$\%I=\frac{Abs\;reference-Abs\;probe}{Abs\;reference}\times 100$$

A curve is plotted % I f(log (Extract Volume)) and the line formula is used to find the IC_50_ value, the antioxidant concentration for which 50% inhibition of the free radicals is observed.

Determining antioxidant activity using the DPPH method was performed according to the method reported by Madhu [133], with minor modifications.

#### 3.3.2. Antioxidant Activity by the CUPRAC Method

The principle underlying the CUPRAC method consists of the reduction in the cupric complex, neocuproine (Cu (II)-Nc), by antioxidants in the cuprous form (Cu (I)-Nc). The reduction of copper ions was performed according to the method described by Celik et al. [134], as follows. A measure of 60 μL of sample/standard solutions of different concentrations was mixed with 50 μL CuCl2 (10 mM), 50 μL neocuproine (7.5 mM), and 50 μL 1 M ammonium acetate buffer, pH = 7.00. The samples were incubated for 30 min, and the absorbance was measured at 450 nm.

#### 3.3.3. Antioxidant Activity by the FRAP Method

The determination of the antioxidant power of reducing iron was performed using the method described by Corbu et al. [135] and Thaipong et al. 2006 [136], with some modifications. The following solutions were prepared: 300 mM acetate buffer, pH 3.6, 10 mM TPTZ (2,4,6-tripyridyltriazine) stock solution in 40 mM HCl, and 20 mM FeCl3 solution in distilled water. The FRAP reagent was prepared by mixing 300 mM acetic acid–sodium acetate buffer solution, pH 3.6, with 10 mM TPTZ solution and 20 mM FeCl_3_ solution (10:1:2). The FRAP reagent was kept in a water bath at 37 °C until the analysis was performed. Over 10 µL sample/standard solution, 190 µL FRAP reagent was added and incubated for 30 min at 37 °C. After incubation, the absorbance was read at 593 nm. A 1 mM Trolox stock solution was used to plot the calibration curve, with concentrations ranging from 30 to 250 μM Trolox/mL.

#### 3.3.4. Antioxidant Activity by TEAC (ABTS) Method

Another commonly used antioxidant capacity assay is the ABTS (2,2’-azino-bis (3-ethylbenzothiazoline-6-sulfonic acid)) method. It too has a mixed mode of functioning, combining HAT and ET mechanisms, although the ET mechanism happens faster. The ABTS^•+^ free radical has a blue-green colour and absorbs at 417, 734, and 815 nm. When reduced by antioxidants, it decreases the solution absorbance at 734 nm. The reaction can thus be followed by spectrophotometry, the decrease in absorbance being proportional to the logarithm of the antioxidant volume. By measuring the absorbance at 734 nm, one limits the interference of other compounds found in the extract and the turbidity. The method is simple, fast (the reaction takes less than 30 min), and can be used on a wide pH range for both lipophilic and hydrophilic compounds in both aqueous and organic solvents. Because the free radicals are not commercially available, 7 mM of ABTS diammonium salt was dissolved and oxidised by 2.5 mM potassium persulfate, in distilled water, to form the oxidant species by the emission of one electron from a nitrogen atom. The solution was left for 16 h to generate the free radicals, turning a dark teal colour. The reactive solution was later diluted in methanol until reaching an absorbance of 0.7 at 734 nm. For the extract, 3 probes were made containing 2.900 mL of the ABTS reactive and 0.100 mL of the methanolic extract solutions of increasing concentrations. After 5 min, the absorbance at 734 nm was measured, and the inhibition percentage was calculated as stated above. A plot %I = f(log extract Volume) was later made for each extract, and the IC_50_ value was found.

### 3.4. Germination Bioassay

This study aimed to evaluate the influence of the extract obtained from fenugreek seeds (*Trigonella foenum-graecum*) in 40% ethanol on wheat seeds. The method used was adapted from the procedures described by Mitelut and Popa [72], Ghayal et al. [137], and Perisoara et al. [138]. The *Trigonella foenum-graecum* extract, as well as the solvent (40% ethanol), were studied at concentrations of 0.10%, 0.50%, 1.00%, and 1.50%, diluted with distilled water, and the results were reported compared to a negative control, in this case, distilled water. Before starting the study, the Petri plates, with a diameter of 100 mm × 200 mm (Corning Petri plates, New York, NY, USA), were disinfected by being sprayed with isopropyl alcohol 70% (Contec IPA) and left to dry overnight. The filter papers (IDL. GMBH, type blue, 90 diameters) were sterilised in a hood equipped with a UV lamp (JOUAN MSC 12, Yerville, France) for 30 min on each side and the wheat seeds (*Triticum aestivum*) were washed with purified water and dried in an incubator (Memmert IPP30, Schwabach, Germany) at a temperature of 30 °C. After completing the washing, sterilisation, and disinfection procedures, 20 wheat seeds (of medium size) were placed in each Petri plate containing the filter paper, over which 5 mL of the extract sample obtained following the dilutions was pipetted. The same procedure is used for testing the negative control and the solvent (40% ethanol), replacing the 5 mL of the extract sample with the negative control/solvent sample. The entire experiment was performed in triplicate, with each plate containing 20 wheat seeds. The Petri plates were incubated in the dark (Memmert IPP30 incubator) for 6 days, at a temperature of 25 ± 1 °C. At the end of the incubation period, germinated seeds were counted, and the length of the roots of the germinated seeds was measured. Based on the results obtained, the following indicators were determined: germination percentage (GP), relative seed germination index (RSG), relative root growth index (RRG), and germination index (Gi), according to the calculation formulas below [79,139]:(2)$$\mathrm{GP}=\frac{\mathrm{Number of germinated seeds}}{\mathrm{Number of total seeds tested}}\times 100$$
(3)$$\mathrm{RSG}=\frac{\mathrm{Average number of germinated seeds in the sample}}{\mathrm{Average number of germinated seeds in the control}}\times 100$$
(4)$$\mathrm{RRG}=\frac{\mathrm{Average root length of germinated seeds in the sample}}{\mathrm{Average root length of germinated seeds in the control}}\times 100$$
Gi = RSG × RRG/100.(5)

According to the literature [76,78], it appears that Gi values, depending on the results obtained, may indicate a phytostimulant or phytotoxic effect of the extract on the seeds studied. For the determination and interpretation of Gi values, the data obtained regarding RRG and RSG were processed [78]. Obtaining a Gi value lower than 50% indicates a strong phytotoxic activity of the extract, a Gi value as close as possible to 0% highlights extreme phytotoxicity, a Gi value between 50 and 80% is associated with moderate phytotoxic activity on the plant, a value above 80% is considered to indicate that the tested extract does not manifest phytotoxic activity, and a Gi value above 100% indicates that the studied extract possesses compounds with phytostimulant properties [76,78]. The experiment ends after 65% of the seeds in the control sample have germinated and/or developed roots of at least 20 mm in length [140,141].

### 3.5. Catalase, Superoxide Dismutase, Peroxidase, and Lipid Peroxidation Assessment

The method used to sterilise the seeds was the incubation of seeds for 30 min at 50 °C in distilled water. The seeds, thus conditioned, were evenly distributed in Petri dishes of 50 mm diameter on a soapy bed of filter paper which had been previously sterilised by UV irradiation and moistened with distilled water. In the case of samples where the simulation of moderate drought conditions was desired, the saddling material was moistened with PEG 6000 to achieve an osmotic potential of −0.6 Mpa [142]. They were covered with new filter paper, and then the test solution was added by pipette (about 10 mL). Three replicates were performed for each species. The control variant was moistened with distilled water. The test solution was dripped in as many places as possible to wet the paper evenly. The Petri dish was covered with the upper half, where the test variant and the repetition were marked with a marker. The plates were covered with aluminium foil and randomly placed in a constant 25 °C environment. Seeds were checked daily to monitor germination progress while ensuring that the filter paper did not dry out. If necessary, a few drops of the test solution were added to maintain optimal humidity. Germination was considered complete when the root became visible. Seeds treated for 6 days were sampled for the determination of biochemical parameters (catalase, superoxide dismutase, peroxidase, and lipid peroxidation). Catalase activity was investigated according to the method of Aebi [143]. The estimation was performed spectrophotometrically, measuring the decrease in absorbance at 240 nm. The reaction mixture contained 0.01M phosphate buffer (pH 7.0), 2 mM H_2_O_2_, and cell lysates. The specific activity of catalase is expressed in terms of units/mg protein. A unit is defined as the velocity constant per second. Superoxide dismutase is a metalloenzyme that catalyses the dismutation of superoxide anion into oxygen and hydrogen peroxide. We have been using the spectrometric procedures described by Sigma Aldrich to determine the SOD activity in samples. The method is based on the spectrophotometric evaluation (550 nm absorption spectra) of the inhibition rate of cytochrome C reduction by competing for the superoxide radical with superoxide dismutase [144]. Peroxidase was analysed according to Prochazkova [145]. The 3 mL reaction mixture contained 28 μL guaiacol, 2 mM H_2_O_2_, 0.1 M phosphate buffer (pH 6.0), and 0.1 mL enzyme. Absorbance was recorded at 470 nm. Lipid peroxidation was evaluated using the method with thiobarbituric acid (TBA). The MDA-TBA adducts formed after the reaction of TBA and MDA from the biological sample were measured using a colorimetric method (λ = 532 nm).

### 3.6. Determination of Antifungal Activity

#### 3.6.1. Qualitative Screening of the Antifungal Activity of the 40% Ethanol-Conditioned Extract of *Trigonella foenum-graecum* L. Seeds Against Some Pathogenic Fungal Species

Qualitative screening enables the collection of preliminary information on the mode of action, the effect of a potential antifungal product (Fenugreek seed extract) on fungal cultures on the semi-solid substrate, as well as the stability of preliminary concentrations in correlation with the effect obtained. Qualitative screening of the antifungal activity of the *Trigonella foenum-graecum* L. seeds in 40% ethanol was performed according to protocols outlined in the specialised literature [146,147,148] and the CLSI M38M51S standard [149], using three pathogenic fungal species, namely *Aspergillus niger* and *Fusarium graminearum* obtained from the Microbial Strains Collection of the Faculty of Biology, University of Bucharest, Romania; and *Monilinia laxa* obtained from the Microbial Strains Collection of the Faculty of Agriculture, University of Agronomic Sciences and Veterinary Medicine Bucharest and confirmed by the MALDI-TOF method.

The evaluation was performed using the disc-diffusion method on the semi-solid PDA (Potato Dextrose Agar) medium, liquefied, cooled to 40–45 °C, and poured into Petri dishes. The culture medium was inoculated with the fungal inoculum adjusted to the nephelometric standard 1 McFarland (106 CFU/mL). Sterilised filter paper discs (6 mm in diameter) were dispersed on the surface of the inoculated plates, over which 10 µL of the extract was prepared at concentrations of 500 mg/mL,250 mg/mL, 125 mg/mL, and 62.5 mg/mL. After 5–7 days of incubation at 27 ± 1 °C, the inhibition diameter of the colonies developed on the culture medium was measured. The diameter of inhibition of colonies formed by fungal strains tested in the presence of Fenugreek extract and the solvent represented by 40% ethanol was compared with that of colonies exposed to the antifungal control (Nystatin, Amphotericin B, Imidazole)—positive control. The negative control (absence of extract and antifungal) was represented by DMSO. The determination of the relative percentage of inhibition (PRI) was performed using the formula [150](6)$$\mathrm{PRI}=\frac{100(X-Y)}{\left(Z-Y\right)}$$
where X = total inhibition area of the extract/solvent tested; Y = total inhibition area of the negative control (DMSO); and Z = total inhibition area of the standard fungicide (positive control). The total inhibition area was calculated using area = πr2, where r = radius of the inhibition zone.

#### 3.6.2. Quantitative Determination of the Antifungal Activity of the *Trigonella foenum-graecum* L. Seeds Extract in 40% Ethanol on Some Pathogenic Fungal Species

The quantitative determination of antifungal activity was performed using the microdilution method according to the protocols in the specialised literature [146,147,148] and the CLSI M38M51S standard [149], using the pathogenic fungal species *Aspergillus niger* and *Fusarium graminearum*. Using RPMI (Roswell Park Memorial Institute) 1640 medium, in 24-well plates, the fungal inoculum, adjusted to the nephelometric standard 1 McFarland (106 CFU/mL), was seeded. The final tested concentrations of the extract (500 mg/mL, 250 mg/mL, 125 mg/mL, and 62.5 mg/mL) were made at a final volume of 2 mL, directly in the culture medium. After 5–7 days of incubation at a temperature of 27 ± 1 °C, the samples were read spectrophotometrically at λ = 620 nm to determine the cell viability (%). The results obtained were compared with the positive control (fungistatic agent) and the negative control (fungal culture without treatment) to determine the antifungal activity, using the following formula, according to Javed et al. [151]:Inhibitory concentration (%) = (A − B)/A × 100(7)
where A is the concentration of the negative control (fungal culture without treatment) and B represents the concentration of the positive control (culture treated with extract/ethanol).

## 4. Statistical Analysis

All studies were performed in triplicate. The results were reported as averages, and the mean standard deviation (±) and relative standard deviation (RSD) were calculated. The statistical analysis for the enzyme activity was performed using Microsoft Excel 2019. Statistical evaluations for the germination bioassay and antifungal activity were performed using GraphPad Prism 9 (San Diego, CA, USA). Data were analysed using the two-way ANOVA test. The level of significance was set to * *p* < 0:05, ** *p* < 0:01, and *** *p* < 0:001 vs. control.

## 5. Conclusions

In this study, we obtained an extract from *Trigonella foenum-graecum* seeds, rich in phytochemical compounds known for their antioxidant and antimicrobial properties, which could be effectively utilised in the plant protection industry.

The extract was found to contain significant levels of reducing sugars; proteins; phenolic compounds, including flavonoids and polyphenols, such as rutin, abscisic acid, salicylic acid, and vitexin; and steroidal saponins, the most notable of which are diosgenin and its precursor, protodiosgenin; as well as a compound belonging to the pyridine alkaloids category, trigonelline. The compounds identified are well documented in the literature for their antimicrobial and antioxidant activities.

Both qualitative and quantitative assessments confirmed the antifungal activity of the extract, particularly against *Fusarium graminearum* and *Monilinia laxa* strains at the highest concentration tested (500 mg/mL). These findings are consistent with the observed phytotoxic effect on wheat seeds at the maximum concentration (1.50%), which can be attributed to the high accumulation of biological compounds in the extract. Future research will include studies on plants under greenhouse conditions to establish effective concentrations for field application, taking into account soil dilution, plant surface absorption, and phytotoxicity thresholds.

This work provides valuable insights into obtaining a potential biologically active product ecologically and efficiently, without a negative impact on the environment, with proven antifungal properties. It also offers a promising alternative for protecting cereal and fruit crops.

## Figures and Tables

**Figure 1 plants-14-03320-f001:**
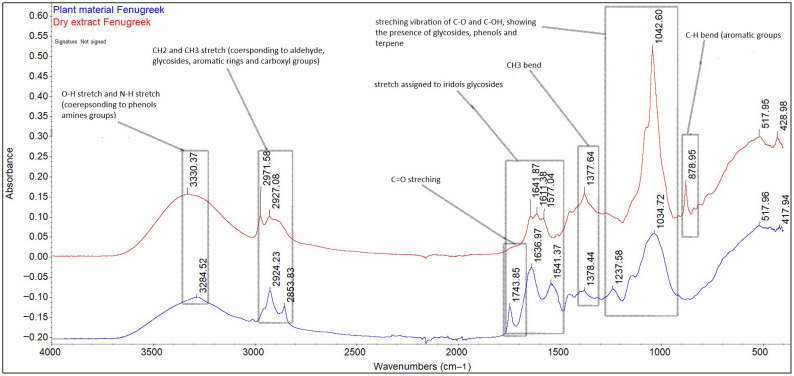
Identification of biological compounds in *Trigonella foenum-graecum* seed extract in 40% ethanol by ATR-FT-IR.

**Figure 2 plants-14-03320-f002:**
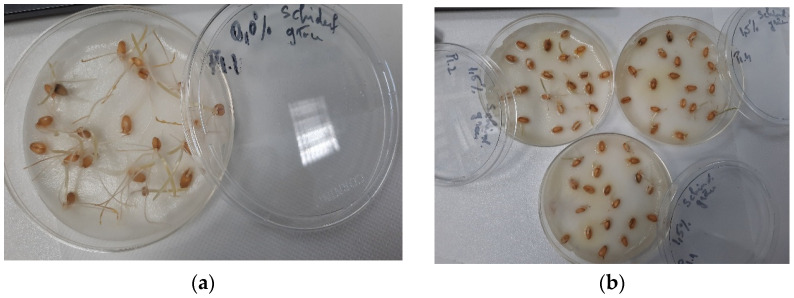
Seedlings growth germination bioassay of wheat seeds using *Trigonellla foenum graecum* seed extract in 40% ethanol: (**a**) control and (**b**) 1.50% extract.

**Figure 3 plants-14-03320-f003:**
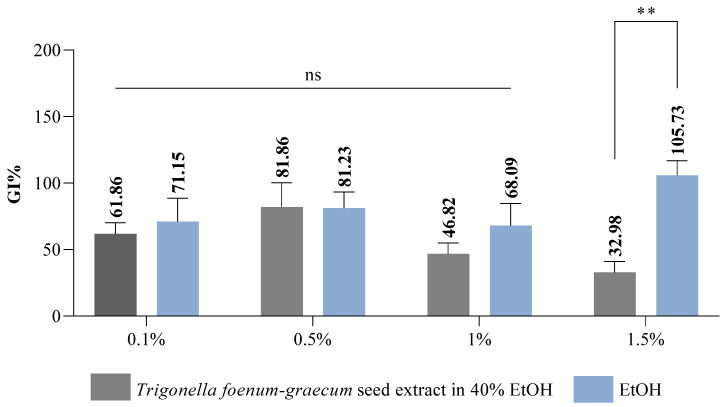
Germination index (Gi) results of *Trigonella foenum-graecum* seed extract variant on wheat seeds; ** *p* value = 0.005 (*n* = 3); ns = non-significant statistically.

**Figure 4 plants-14-03320-f004:**
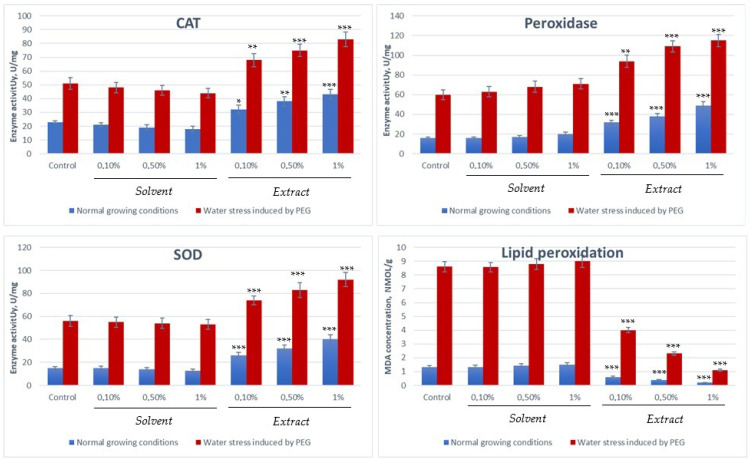
Evaluation of the enzyme activity of enzymes involved in counteracting oxidative stress and the degree of lipid peroxidation (* *p* < 0:05, ** *p* < 0:01 and *** *p* < 0:001 vs. control).

**Figure 5 plants-14-03320-f005:**
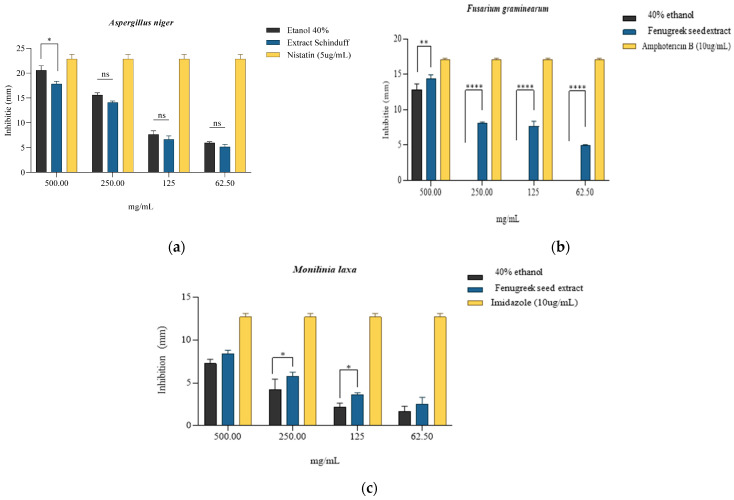
Qualitative antifungal activity expressed by inhibition zones (mm) of the extract obtained from *Trigonella foenum-graecum* L. seeds, with 40% ethanol used as solvent and specific fungistatic controls for strains of (**a**) *Aspergillus niger*, * is *p* = 0.0274; (**b**) *Fusarium graminearum*, ** is *p* = 0.0037, **** = *p* < 0.0001; and (**c**) *Monilinia laxa*, * is *p* = 0.0455; ns = non-significant statistically; *n* = 3.

**Figure 6 plants-14-03320-f006:**
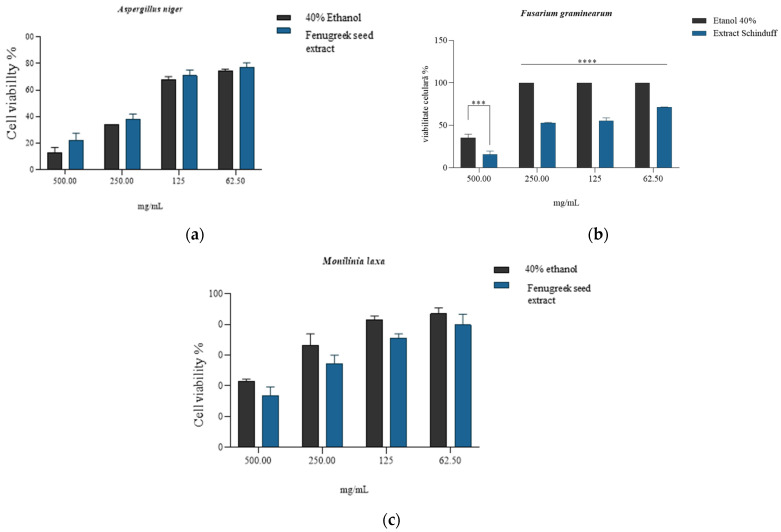
Quantitative antifungal activity of the *Trigonella foenum-graecum* L. (Fenugreek) seed extract in 40% ethanol, compared to that obtained for solvent (40% ethanol) on strains of *Aspergillus niger* (**a**), *Fusarium graminearum* (**b**), and *Monilinia laxa* (**c**), where *** is *p = 0.0002*, **** = *p* <0.0001; ns = non-significant statistically; *n* = 3.

**Figure 7 plants-14-03320-f007:**
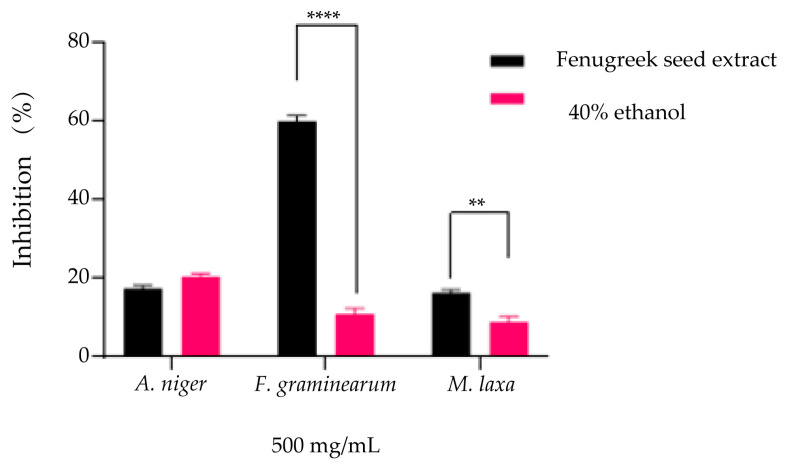
Evaluation of the antifungal effect by inhibition of mycelium growth (%) on PDA medium compared to the positive control (untreated strain) for both the *T. foenum-graecum* seed extract and the solvent (40% ethanol) used at similar concentrations (** *p* = 0.0016, **** *p* = < 0.0001) on the strains *Aspergillus niger*, *Fusarium graminearum*, and *Monilinia laxa*.

**Table 1 plants-14-03320-t001:** Phenolic compounds content found in the *Trigonella foenum-graecum* seed extract in 40% ethanol (* mean ± SD, *n* = 3).

Parameter	*Trigonella foenum-graecum* Extract
TFC—Rutin mg/L	3.9517 ± 0.007 *
TPC—caffeic acid mg/L	36.817 ± 0.65 *

**Table 2 plants-14-03320-t002:** The qualitative content of the biological compounds present in *Trigonella foenum-graecum* seed extract was determined by FT-ICR-MS.

Compound	Chemical Formula	(*m*/*z*) Measured	(*m*/*z*) Generated
Diosgenin	C_27_H_42_O_3_	415.32089	415.32067
Trigonelline	C_7_H_7_O_2_	138.05504	138.05495
Tigogenin	C_27_H_44_O_3_	4.7.33657	417.33632
Yamogenin	C_27_H_42_O_3_	415.32089	415.32067
Vitexin	C_21_H_20_O_10_	433.11315	433.11292
Acid salicylic	C_7_H_6_O_3_	139.03901	139.03897
Apigenin 6-C-galactoside 8-C—arabinoside	C_26_H_28_O_14_	565.11543	565.11518
Luteoline-7-O-glucoside	C_21_H_18_O_12_	463.08932	463.08710
Apigenin-7-O-glucoside	C_21_H_20_O_10_	433.11315	433.11292
Protodiosgenin	C_51_H_84_O_22_	1049.55408	1049.55270
Acid abscisic	C_15_H_20_O_4_	265.14352	265.14344
Rutin	C_27_H_30_O_16_	611.16110	611.16066
t-Resveratrol	C_14_H_12_O_3_	228.07554	228.07810

**Table 3 plants-14-03320-t003:** Reducing sugar and soluble protein content found in *Trigonella foenum-graecum* seed extract in 40% ethanol (* mean ± SD, *n* = 3).

Parameter	*Trigonella foenum-graecum* Seed Extract
Total reducing sugar (mg D-glucose/mL)	19.20 ± 0.35 *
Total soluble proteins (mg BSA/mL)	1.96 ± 0.07 *

**Table 4 plants-14-03320-t004:** Antioxidant activity evaluation of *Trigonella foenum-graecum* extract.

Parameter	Volume of Extract Required to Reduce the Level of Reactive Species with 50% (μL)
DPPH	2.934 ± 0.023 *
ABTS	3.011 ± 0.013 *
FRAP	5.29 ± 0.34 *
CUPRAC	11.64 ± 0.64 *

* mean ± SD, *n* = 3.

**Table 5 plants-14-03320-t005:** Results of germination percentage (GP%) of fenugreek seed extract and solvent (40% ethanol) (* mean ± SD, *n* = 3).

Sample Name	% Sample				Control
	0.10	0.50	1.00	1.50	
Fenugreek seed extract conditioned in 40% ethanol	91.67 ± 0.57 * RSD = 3.14	91.67 ± 2.08 * RSD = 11.35	88.33 ± 2.51 * RSD = 14.54	85 ± 0.66 * RSD = 5.88	96.67 ± 1.15 * RSD = 5.97
40% Ethanol (solvent)	91.66 ± 2.09 * RSD = 11.35	81.66 ± 2.51 * RSD =15.40	73.33 ± 1.52 * RSD = 10.41	90 ± 2.47 * RSD = 11.11	93.33 ± 1.15 * RSD = 6.18

**Table 6 plants-14-03320-t006:** Results of the relative germination index (RSG%) of fenugreek seed extract and solvent (40% ethanol).

Sample Mean	% Sample				Control
	0.10	0.50	1.00	1.50	
Fenugreek seed extract conditioned in 40% ethanol	94.8 **	94.82 **	91.37 **	87.93 **	96.67 ± 1.15 RSD = 5.97
40% Ethanol (solvent)	98.21	87.5	78.57	96.42	93.33 ± 1.15 RSD = 6.18

+ statistically significant *p* < 0.05, compared to the control sample; ++ statistically significant *p* < 0.05 compared to the solvent. *n* = 3, ** ns = non-significant statistically.

**Table 7 plants-14-03320-t007:** The relative root growth index results (RRG%) following the treatment of wheat seeds with fenugreek seed extract and solvent (40% ethanol); mean ± SD, (*n* = 3).

Sample Name	% Sample				Control
	0.10	0.50	1.00	1.50	
Fenugreek seed extract conditioned in 40% ethanol	+ 64.29% (1.19 ± 0.20 RSD = 17.24)	85.70% (1.58 ± 0.56 RSD = 35.43)	+ 50.00 (0.92 ± 0.14 RSD = 15.20)	+;++ 35.79% (0.66 ± 0.14 RSD = 22.11)	* 1.85 ± 0.17 RSD = 9.22
40% Ethanol (solvent)	70.48% (1.6 ± 0.47 RSD = 29.81)	91.18% (2.07 ± 0.23 RSD = 11.13)	84.87% (1.92 ± 0.71 RSD = 36.93)	109.10% (2.47 ± 0.47 RSD = 19.31)	2.27 ± 0.36 RSD = 16.08

* root length is measured in cm; + statistically significant *p* < 0.05, compared to the control sample; ++ statistically significant *p* < 0.05 compared to the solvent.

**Table 8 plants-14-03320-t008:** Comparative qualitative evaluation of antifungal activity for different concentrations of *Trigonella foenum-graecum* L. seed extract in 40% ethanol against the phytopathogenic strains *Aspergillus niger*, *Fusarium graminearum*, and *Monilinia laxa* (mean ± SD, *n* = 3).

Concentrations (mg/mL)	Extract	*Aspergillus niger*	*Fusarium graminearum*	*Monilinia laxa*
		Inhibition Percentage (%)	Inhibition Percentage (%)	Inhibition Percentage (%)
500	Fenugreek seed extract conditioned in 40% ethanol	78.01 ± 5.61	84.43 ± 4.29	66.23 ± 5.55
	40% ethanol (solvent)	87.32 ± 5.74	64.56 ± 5.88	57.05 ± 1.99
	*p* value	<0.05	<0.05	<0.05
250	Fenugreek seed extract conditioned in 40% ethanol	61.78 ± 3.72	47.21 ± 0.66	45.37 ± 5.41
	40% ethanol	66.24 ± 0.41	0.00 ± 0.00	33.64 ± 10.59
	*p* value	<0.05	<0.05	<0.05
125	Fenugreek seed extract conditioned in 40% ethanol	37.80 ± 1.54	44.82 ± 3.63	28.78 ± 2.63
	40% ethanol	32.50 ± 3.72	0.00 ± 0.00	16.87 ± 3.33
	*p* value	<0.05	<0.05	<0.05
62.5	Fenugreek seed extract conditioned in 40% ethanol	22.62 ± 3.11	28.77 ± 0.51	20.19 ± 6.80
	40% ethanol	25.30 ± 1.46	0.00 ± 0.00	13.08 ± 5.45
	*p* value	<0.05	<0.05	<0.05

## Data Availability

The data presented in this study and supplymentary material and are available in article.

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
