# Peer review of "In Vitro Evaluation of the Antifungal Activity of Trigonella foenum-graecum Seed Extract and Its Potential Application in Plant Protection"

_plants, 2025, doi:10.3390/plants14213320_

Round 1

Reviewer 1 Report

Comments and Suggestions for Authors

This manuscript presents interesting information and is well written, presented, and discussed. However, some minor revisions should be made before it can be accepted for publication in this journal.

Lines 35-36 This line should be rewitten.

Some studies on the use of Trigonella foenum-graecum seed extracts in aother similar studies should be included.

Spectrum Evaluation by ATR-FT-IR: Disclose the importance to carried ot this assay, as FT-ICR-MS as well.

Author Response

Reviewer 1

Dear Editor in Chief,

Dear Reviewer,

We would like to thank you for your observations and the effort that you made to improve our manuscript. We made all the proposed corrections. Please check our responses.

Comments and Suggestions for Authors

This manuscript presents interesting information and is well written, presented, and discussed. However, some minor revisions should be made before it can be accepted for publication in this journal.

Lines 35-36 This line should be rewritten.

Replay: We rephrased (now lines 34-39).

Some studies on the use of Trigonella foenum-graecum seed extracts in another similar studies should be included.

Replay: Thank you for your observation. We have included similar studies. These can be found both in the study of phytochemical compounds - TPC, at lines 176-187 (formerly lines 170-180); TFC, at lines 194-197 (formerly lines 196-199); determination of phytochemical compounds by the FT-ICR-MS ultra-high resolution and precision of the mass-to-charge ratio (m/z) method, lines 224-238; determination of the content of soluble proteins and carbohydrates, lines 268-279 (formerly lines 261-273)

We also included studies lines 326-332, 389-397, We also inserted new references [73,85] (new version)

Spectrum Evaluation by ATR-FT-IR: Disclose the importance to carried out this assay, as FT-ICR-MS as well.

Replay: The importance of using the FT-IR technique through ATR is to quickly identify compounds or classes of compounds that are part of the obtained plant extract. It is a non-destructive method, providing us with useful information about the quality of the extract and the structure of the organic compounds in the sample in a short time. The identification of specific bands was explained between lines 127-155 (new version). It is also worth mentioning that some bands specific to the compounds of interest were found/transferred in the obtained dry extract.

FT-ICR (Fourier Transform Ion Cyclotron Resonance) is a high-resolution mass spectrometry technique that determines the mass-to-charge ratio (m/z) of ions by measuring their cyclotron frequency in a strong magnetic field. This method is known for its unparalleled accuracy and resolution, being used to accurately determine the monoisotopic mass of the compound of interest, with a precision of up to 5 decimal places. These are explained in the text between lines 225-255 (new version).

FT-IR analysis gives us information about the classes of compounds present in the extract, and FT-ICR comes by identifying the compounds (by accurately measuring the molecular mass between 5-7 decimal places, a precision that cannot be achieved with any other type of mass spectrometry) that exhibit the antifungal effect on fungal cultures. We mention that it is important to know the identity of the compounds with this effect, in order to highlight the antifungal effect of the extract obtained from fenugreek seeds.

Reviewer 2 Report

Comments and Suggestions for Authors

The manuscript presents a comprehensive study on the in vitro antifungal potential of Trigonella foenum-graecum (fenugreek) seed extract as a sustainable alternative to chemical fungicides. The topic is highly relevant, aligning with global efforts toward eco-friendly plant protection strategies.
The experimental design is generally robust, combining phytochemical profiling (ATR-FT-IR, FT-ICR-MS), antioxidant evaluation, phytotoxicity assays, and antifungal testing. The integration of biochemical and biological data is a major strength.
Overall, the paper demonstrates scientific rigor and novelty, though some methodological and interpretational clarifications are needed.
1. Strengths
•    Innovative scope: The manuscript connects secondary metabolite profiling with in vitro antifungal activity, offering potential for biofungicide development.
•    Comprehensive analytical approach: Use of FT-ICR-MS and ATR-FT-IR is commendable and provides high-quality phytochemical identification.
•    Ecological relevance: The study supports the transition to sustainable, non-toxic plant protection methods.
•    Clarity of results: The antifungal data are systematically presented with appropriate figures and tables.
2. Major Comments
1.    Rationale for Concentration Range: The choice of extract concentrations (62.5–500 mg/mL) needs clearer justification. Are these concentrations physiologically or agronomically relevant? Clarify whether these doses correspond to potential field applications or were chosen for screening purposes only.
2.    Phytotoxicity Discussion: The observed inhibitory effects on wheat seed germination at 1.5% concentration are significant. Expand on possible mechanisms of this phytotoxicity—e.g., phenolic-induced oxidative stress, interference with hormone balance, or osmotic effects.
3.    Fungal Specificity: The extract exhibited strong activity against Fusarium graminearum but weaker effects against Aspergillus niger and Monilinia laxa. Please discuss potential differences in cell wall composition or target mechanisms that may explain this selectivity.

Author Response

Reviewer 2

Dear Editor in Chief,

Dear Reviewer,

We would like to thank you for your observations and the effort that you made to improve this manuscript. We made all the proposed corrections. Please check our responses.

Comments and Suggestions for Authors

The manuscript presents a comprehensive study on the in vitro antifungal potential of Trigonella foenum-graecum (fenugreek) seed extract as a sustainable alternative to chemical fungicides. The topic is highly relevant, aligning with global efforts toward eco-friendly plant protection strategies.

The experimental design is generally robust, combining phytochemical profiling (ATR-FT-IR, FT-ICR-MS), antioxidant evaluation, phytotoxicity assays, and antifungal testing. The integration of biochemical and biological data is a major strength.

Overall, the paper demonstrates scientific rigor and novelty, though some methodological and interpretational clarifications are needed.

  1. Strengths
  • Innovative scope: The manuscript connects secondary metabolite profiling with in vitro antifungal activity, offering potential for biofungicide development.
  • Comprehensive analytical approach: Use of FT-ICR-MS and ATR-FT-IR is commendable and provides high-quality phytochemical identification.
  • Ecological relevance: The study supports the transition to sustainable, non-toxic plant protection methods.
  • Clarity of results: The antifungal data are systematically presented with appropriate figures and tables.
  1. Major Comments
  2. Rationale for Concentration Range: The choice of extract concentrations (62.5–500 mg/mL) needs clearer justification. Are these concentrations physiologically or agronomically relevant? Clarify whether these doses correspond to potential field applications or were chosen for screening purposes only.

Response: We thank the reviewer for this valuable observation. The concentrations used in this study (62.5–500 mg/mL) were selected for in vitro screening purposes to determine the range of antifungal activity and minimum inhibitory concentration (MIC). These values do not directly correspond to agronomic doses but rather serve as a preliminary assessment of the bioactive potential of Trigonella foenum-graecum extract under controlled laboratory conditions (lines 554-562, new version)

In subsequent research, we plan to perform in planta and greenhouse trials to establish effective concentrations for field application, taking into account soil dilution, plant surface absorption, and phytotoxicity thresholds.

  1. Phytotoxicity Discussion: The observed inhibitory effects on wheat seed germination at 1.5% concentration are significant. Expand on possible mechanisms of this phytotoxicity—e.g., phenolic-induced oxidative stress, interference with hormone balance, or osmotic effects.

Response: We thank the reviewer’s observation. We have included new phrases to emphasize the impact of phenolic compounds on germinated seeds when found in high concentrations, now line 483-491 (new version). We also inserted new references [93,94] (new version)

  1. Fungal Specificity: The extract exhibited strong activity against Fusarium graminearum but weaker effects against Aspergillus niger and Monilinia laxa. Please discuss potential differences in cell wall composition or target mechanisms that may explain this selectivity.

Response: We appreciate the reviewer’s insightful comment. The observed differential sensitivity among the tested fungal species can likely be attributed to structural and biochemical variations in their cell walls and membrane composition. Fusarium graminearum possesses a relatively thinner chitin-glucan matrix and higher ergosterol turnover, making it more susceptible to membrane-disrupting compounds such as steroidal saponins and phenolic acids. In contrast, Aspergillus niger and Monilinia laxa have more melanised or heavily cross-linked cell walls that enhance rigidity and resistance to oxidative or surfactant-type stress, potentially reducing extract penetration and activity. Moreover, the extract’s main bioactive constituents — trigonelline, diosgenin, and flavonoids — act through oxidative imbalance and membrane permeability disruption, mechanisms that may be more effective against Fusarium species than against heavily melanised or spore-forming fungi such as Aspergillus. (Rodríguez-Pires et al. 2020; Chhoker, Hausner, and Harris 2025). (lines 690 – 700, new version)

We also inserted new references [121,122,123] (new version)

Reviewer 3 Report

Comments and Suggestions for Authors

This study, titled "In Vitro Evaluation of the Antifungal Activity of Trigonella foenum-graecum Seed Extract and Its Potential Application in Plant Protection," examines in detail the antifungal, antioxidant, and phytotoxic properties of Trigonella foenum-graecum (fenugreek) seed extract. The study addresses a current and relevant topic that could contribute to the development of environmentally friendly biofungicides for plant protection. The experimental design is generally strong, and the methods used are clear and reproducible. However, the discussion sections are superficial in some places, and the relationship between the results and the biochemical components is not sufficiently explored. Furthermore, there are deficiencies in the data presentation and statistical analyses. While the English language is generally understandable, language editing is needed to improve scientific style and fluency. Overall, the article could be made suitable for publication with comprehensive scientific and structural revisions.

There are some areas for improvement in the article. My suggestions for this are listed below.

1- Although the introduction section comprehensively summarizes the literature, the difference and scientific contribution of this study from previous studies on Trigonella foenum-graecum or similar plant extracts are not clearly emphasized. The novelty of the study should be more clearly stated.

2- The methodology section states that the extraction was first performed with 70% ethanol and then continued with 40% ethanol. The relationship and purpose between these two stages should be clearly explained.

3- Although the use of an ANOVA test is stated, statistical significance indicators (p-values, letter groups, etc.) are not shown in the tables and graphs. The number of replicates (n) for each analysis should be clearly stated.

4- The phytochemical components are described in detail, but their relationship to the antifungal mechanism of action is not sufficiently explained. For example, the effects of saponins or flavonoids on cell wall permeability should be discussed with literature support.

5- Tests conducted on wheat seeds are detailed, but the dose-response relationship is not quantitatively demonstrated. Additionally, adding visual support (photographs or graphs) will strengthen the results.

6- Some figures (especially enzyme activity and antifungal test results) do not include error bars or standard deviation values. Units of measurement (mg/mL, %, IC₅₀, etc.) should be standardized.

7- Although percentage inhibition rates are given, Minimum Inhibitory Concentration (MIC) values ​​are not clearly stated. The distinction between "fungicidal" and "fungistatic" effects should be supported by experimental evidence.

8- Some figure legends (e.g., Figures 4–6) are missing, or abbreviations are not defined. The resolution and statistical indicators of the figures should be improved.

9- The English text contains long, repetitive sentences. The language should be simplified to maintain a scientific and academic tone, and unnecessary repetitions should be removed.

10- The Results section focuses on repetition of findings. Instead, The study's potential applications (biofungicide formulation, field trials, etc.), limitations, and future research directions should be highlighted.

11- It is also important to use scientific names in this article. Please use all Latin names correctly. Especially when the species is first used, add the long and author name (Trigonella foenum-graecum L.), then shorten it (T. foenum-graecum). Use this for all Latin names, such as Fusarium graminearum (after the first use, F. graminearum).

Author Response

Reviewer 3

Dear Editor in Chief,

Dear Reviewer,

We would like to thank you for your observations and the effort that you made to improve this manuscript. We made all the proposed corrections. Please check our responses.

Comments and Suggestions for Authors

This study, titled "In Vitro Evaluation of the Antifungal Activity of Trigonella foenum-graecum Seed Extract and Its Potential Application in Plant Protection," examines in detail the antifungal, antioxidant, and phytotoxic properties of Trigonella foenum-graecum (fenugreek) seed extract. The study addresses a current and relevant topic that could contribute to the development of environmentally friendly biofungicides for plant protection. The experimental design is generally strong, and the methods used are clear and reproducible. However, the discussion sections are superficial in some places, and the relationship between the results and the biochemical components is not sufficiently explored. Furthermore, there are deficiencies in the data presentation and statistical analyses. While the English language is generally understandable, language editing is needed to improve scientific style and fluency. Overall, the article could be made suitable for publication with comprehensive scientific and structural revisions.

There are some areas for improvement in the article. My suggestions for this are listed below.

1- Although the introduction section comprehensively summarizes the literature, the difference and scientific contribution of this study from previous studies on Trigonella foenum-graecum or similar plant extracts are not clearly emphasized. The novelty of the study should be more clearly stated.

Reply: Thank you for your comments. We have modified it according to your suggestions, lines 105-114, 116-117 (new version).

2- The methodology section states that the extraction was first performed with 70% ethanol and then continued with 40% ethanol. The relationship and purpose between these two stages should be clearly explained.

Reply: Thank you for your suggestion! One of the main purposes of using this extraction method is: 1 -  to obtain a potential product that could replace chemicals used in plant protection in an ecological way, without high energy consumption, and -  to extract compounds with potential antimicrobial effect in the highest possible concentrations, eliminating the risk of their degradation. As explained in the method of obtaining, the extract obtained by maceration is analysed and re-extracted in 40% ethanol to achieve an extract with a minimum concentration of 5 mg GAE/mL – m/v. 40% ethanol was used to eliminate a possible negative impact on plants when they are treated with the final product. We introduced new lines to explain the motivation of selected solvent concentrations, lines 109-114, 116-117 and 1025-1028 (new version).

3- Although the use of an ANOVA test is stated, statistical significance indicators (p-values, letter groups, etc.) are not shown in the tables and graphs. The number of replicates (n) for each analysis should be clearly stated.

Reply: Thank you for your suggestion! This is found in the material and is highlighted in the table where there was a statistically significant difference. See Table 7. The number of replications is described in the analysis method, line 902 (new version) and in the section Statistical analysis.

We have modified it according to your suggestions: table 8, table 7, table 6, table 5, table 4, table 3, table 1, figure 4 (new version), and lines 411-412, 415-416 (new version).

4- The phytochemical components are described in detail, but their relationship to the antifungal mechanism of action is not sufficiently explained. For example, the effects of saponins or flavonoids on cell wall permeability should be discussed with literature support.

Reply: We thank the reviewer for this valuable comment. We have now expanded the Discussion section to include a detailed explanation of how the main phytochemical groups identified (particularly saponins and flavonoids) contribute to antifungal activity. Specifically, we have described their mechanisms of action related to fungal cell wall and membrane permeability, inhibition of ergosterol biosynthesis, and oxidative stress modulation, supported by recent literature. (lines 238-241 and 675-689). We also inserted new references [115-120] (new version).

5- Tests conducted on wheat seeds are detailed, but the dose-response relationship is not quantitatively demonstrated. Additionally, adding visual support (photographs or graphs) will strengthen the results.

Reply: Thank you for the comment! This is found in the material and is also highlighted by the correlation with other studies conducted by other researchers, lines 467-483 (new version).

Pictures have been added to support the biogerminative study, now Figure 2 (new version).

6- Some figures (especially enzyme activity and antifungal test results) do not include error bars or standard deviation values. Units of measurement (mg/mL, %, IC₅₀, etc.) should be standardised.

Reply: All figures representing enzyme activity and antifungal assays have been revised to include error bars corresponding to the standard deviation (SD) of three independent replicates (n = 3), Figure 4 (new version). The statistical variability is now indicated both visually and numerically in the figure legends. Additionally, units of measurement throughout the manuscript have been standardised according to Plants journal guidelines (mg·mL⁻¹ for concentration, µL for volume, %, and IC₅₀ in µg·mL⁻¹). These corrections have been implemented consistently across the Results section and described in the revised figure legends.

Where it was permitted, we have standardized the units of measurement, but unfortunately we are not able to do this for all of the analytical methods applied because, for example, when dosing phenolic compounds we have mg/L because we dose an existing compound in a volume of extract, for antifungal tests we use % because we prepared a solution containing x% extract and treated the seeds with it.

7- Although percentage inhibition rates are given, Minimum Inhibitory Concentration (MIC) values ​​are not clearly stated. The distinction between "fungicidal" and "fungistatic" effects should be supported by experimental evidence.

Reply: Response: We appreciate this important observation. We have now revised the Results (Section 2.4) to explicitly report the Minimum Inhibitory Concentration (MIC) values determined for each tested fungal strain. The MIC was defined as the lowest extract concentration at which no visible mycelial growth was observed compared with the solvent control. We have also clarified the distinction between fungistatic and fungicidal effects based on the regrowth capacity of fungal mycelia on fresh PDA plates without extract. These methodological details and results have been added to the revised manuscript (lines 630-642). We also inserted new references [114,115-123] (new version)

8- Some figure legends (e.g., Figures 4–6) are missing, or abbreviations are not defined. The resolution and statistical indicators of the figures should be improved.

Reply: Response: We thank the reviewer for highlighting this issue. The figure legends for Figures 5–7 (new version) have been revised and expanded to include full descriptions of the experimental conditions, abbreviations, and statistical indicators. All abbreviations (e.g., MIC, PDA, SD, ns, p values) are now defined in the captions. These corrections have been applied in the revised version of the manuscript.

9- The English text contains long, repetitive sentences. The language should be simplified to maintain a scientific and academic tone, and unnecessary repetitions should be removed.

Response: We apologize! We made the corrections in the text.

10- The Results section focuses on repetition of findings. Instead, The study's potential applications (biofungicide formulation, field trials, etc.), limitations, and future research directions should be highlighted.

Reply: Thank you for the suggestions, but in the results part, only the relevant results are highlighted and on which there are substantial changes. We tried to highlight a justification of the results reported with the specialized literature. And we add future research that we plan to conduct in the section Conclusion (line 1025-1028, new version). And we include new phrases, lines 561-563 (new version)

11- It is also important to use scientific names in this article. Please use all Latin names correctly. Especially when the species is first used, add the long and author name (Trigonella foenum-graecum L.), then shorten it (T. foenum-graecum). Use this for all Latin names, such as Fusarium graminearum (after the first use, F. graminearum).

Replay: We apologise! We made the corrections in the text. 

Round 2

Reviewer 3 Report

Comments and Suggestions for Authors

The authors have made the corrections I suggested in the first version. The article can be published in its updated form. However, the English version contains numerous grammatical errors. It should be reviewed by a native English speaker. It is then suitable for scientific publication.

Comments on the Quality of English Language

the English version contains numerous grammatical errors. It should be reviewed by a native English speaker. It is then suitable for scientific publication.